# Antennal Sensilla Morphology and Flagellomere Addition in Nymphs and Adults of *Hierodula patellifera* Serville, 1839 (Mantodea: Mantidae)

**DOI:** 10.3390/insects16070655

**Published:** 2025-06-24

**Authors:** Huan Wang, Yang Wang, Yang Liu, Geer Teni, Huiwen Li

**Affiliations:** 1Key Laboratory of Resource Biology and Biotechnology in Western China (Ministry of Education) and College of Life Science, Northwest University, Xi’an 710069, China; wh199911232024@163.com (H.W.); m18504793182@163.com (G.T.); 13473131509@163.com (H.L.); 2Shangluo Research Center of Chinese Medicinal Materials Integrated Pest Management, College of Biology Pharmacy and Food Engineering, Shangluo University, Shangluo 726000, China; 3Department of Entomology, University of Manitoba, Winnipeg, MB R3T 2N2, Canada

**Keywords:** antennal development, antennae, SEM, ultrastructure, sexual dimorphism

## Abstract

To better understand the morphology of the antennae of mantids, the types of antennal sensilla and their distributional features on the antennae of each instar nymph and adult of *Hierodula patellifera* Serville, 1839 (Mantodea: Mantidae) were investigated. The antennal morphology of nymphs and adults was similar and consisted of the scape, pedicel, and flagellum. Seven types and four subtypes of sensilla were identified in nymphs and adults. Based on the external morphology and distributional features of the sensilla, we proposed to divide the adult flagellum into six parts. Among the different instar nymphs and adults, the most obvious individual growth of antennae occurred from the 5th- to 6th-instar nymphs; sexual dimorphism in antennal length, width, and sensilla distribution becomes evident from the 7th instar, and compared to adult females, the adult males have larger flagellomeres and a large number of grooved peg sensilla. We observed that the antennae of *H. patellifera* develop with the addition of new flagellomeres at their proximal and secondary division sites.

## 1. Introduction

Insect antennae play an important role in the process of information exchange, such as host location, host discrimination, habitat searching, avoiding danger, mating and reproduction, and pheromone acceptance [1]. The antennae is one of the principal sensory organs in insects for olfactory perception, with various types of sensilla distributed on its surface [2]. Antennae may vary considerably in length, morphology, type, number of segmentation, and size in different sexes of the same species [3]. For most insects, sexual dimorphism in antennae is a ubiquitous phenomenon. In general, the antennae of males are more complex than females because they need more sensilla to detect the sex pheromones of females in the searching process [4,5]. Therefore, the study of insect sensilla is helpful in revealing antennal dimorphism.

Praying mantids are skilled predators that feed on various arthropods and small vertebrates [6]. This diverse group plays an important role in the ecosystem as all mantid species are predatory [7]. Praying mantids depend on acute vision to detect and attack their prey [8]. Because of this characteristic, praying mantids have been the focus of excellent research on vision [9]. However, mantids also use their antennal sensilla for foraging and courtship [10]. In addition, their antennae present atypical features that might help to better understand the olfactory system [11].

Numerous studies have been conducted to examine the sexual dimorphism of antennae in Mantodea insects. Males of false garden mantids *Pseudomantis albofimbriata* Carl Stål, 1860 were able to call females more quickly if the density of their antennal sensilla trichoidea was greater [12]. Grooved peg sensilla are significantly more numerous in male antennae compared to female mantids. This phenomenon helps males to detect sex pheromones [11,13]. In addition, the sensillar distribution is not uniform along the flagellum in adults and varies within and between the sexes [11]. Because of this atypical sensillar distribution, Carle et al. first developed a new and innovative development model to reconstruct mantid antennae by analyzing the length of flagellomeres [14]. However, studies on the addition of antennal flagellomere in mantids remain scarce.

*H. patellifera* is an important and widely distributed predator. Many species of Hierodulinae share common traits in external morphology, resulting in the early misplacement of *Hierodula* and *Rhombodera* [15,16]. To make it easier to distinguish its intraspecific variation, in this study, we selected *H. patellifera* to analyze the types, size, and distribution of the antennal sensilla in adults and nymphs using scanning electron microscopy. We discussed these features between *H. patellifera* and the antennae of other insects. In addition, we observed the addition of new segments at two distinct sites in *H. patellifera*. Our findings contributed to understanding mantid taxonomy and evolution, which could also further reveal the morphological diversity of mantid antennae.

## 2. Materials and Methods

### 2.1. Insect Collection and Maintenance

In April 2022, we collected five *H. patellifera* oothecae from dry branches along a road in Dongwan, Guangdong Province, China. The oothecae were then incubated at a temperature of 26 °C in the Laboratory of Pest and Disease Control, Shangluo University. We raised around 100 neonate nymphs individually in plastic boxes (6 × 6 × 4.5 cm) at ca. 85 ± 1% relative humidity and a photoperiod of 12 L:12 D until the 6th nymphal instar. The 6th and subsequent nymphal instars (including adults) were placed in 8 × 8 × 11 cm insect-rearing containers. We recognized nine nymphal instars in *H. patellifera*. *H. patellifera* were fed on fruit flies (1st–3rd-instar nymphs) and cockroaches (after the 3rd instar) every two days. From the 4th instar, the gender of nymphs can be distinguished by their genitalia.

### 2.2. Sample Preparation for SEM

We anesthetized *H. patellifera* adults and nymphs (five male and female individuals of each instar and adult). The antennae were carefully removed from the head under a dissecting microscope (SMZ745T, Nikon Precision Shanghai Co., Ltd., Shanghai, China) and washed with an ultrasonic cleaner (KQ-250DM, KunShan Ultrasonic Instruments Co., Ltd., Kunshan, China). The cuticular structures of young nymphs are more delicate than the adults and old mature nymphs, so the duration of washing varied depending on the instar. Nymphs at the 1st to 5th instars should be washed at 45 KHz for no more than 2 min, while those after the 5th instars need to be washed at the same frequency for 3 min. After washing, the antennae were dehydrated in an ascending series of ethyl alcohol solutions (30–90%) and, finally, twice with 100% ethanol (15–20 min each). Samples were then placed in a critical point dryer (SCD-350M, ShiAnJia Biotechnology Co., Ltd., Beijing, China) for twelve hours until all samples were totally dry. The dried samples were mounted on aluminum stubs with double-sided copper sticky tape and then sputter-coated with gold for 1 min (GVC-1000, BeJing Gevee-tech Co., Ltd., Beijing, China). Finally, they were examined and photographed using a scanning electron microscope (SEM-3200, CIQTEK Co., Ltd., Hefei, China; SEM-ZEISS Sigma 560, Carl ZEISS Co., Ltd., Shanghai, China) with an accelerating voltage of 7–10 kV.

### 2.3. Statistical Analyses

The length and width of the antennae were measured using the SEM’s built-in software. Images were processed using Adobe Photoshop CC 2020 (Adobe Systems, San Jose, CA, USA) and were created using Origin Pro 2021 (OriginLab, Microcal Company, Northampton, MA, USA). Shapiro–Wilk tests were used to test the normality of the data, and Levene tests were used to determine the homogeneity of variance. Data conforming to the normal distribution and showing homogeneity of variance were analyzed using one-way analysis of variance, and Tukey’s method was used for multiple comparisons. The experimental data were analyzed using the SPSS Statistics 27 package (IBM, Armonk, NY, USA). The results are expressed as the mean ± SE; *p*-values less than 0.05 indicate a significant difference.

### 2.4. Terminology

The identification of various types of sensilla is based on the external morphology, length, and distribution [17,18]. The terminology for the antennal sensilla follows that of Carle et al. and Drilling [2,19]. The abbreviations for the sensilla are as follows: sensilla trichodea: St; sensilla basiconica: Sb; sensilla chaetica: Sc; sensilla coelocapitula: Sco; sensilla campaniformia: Sca; grooved peg sensilla: Sgp; Böhm’s bristle: Bb. In addition, we develop schematics of the addition of flagellomere. This model uses the number 5 to represent the number of flagellomeres in each part from Parts II to VI, illustrating the growth patterns of two growth sites.

## 3. Results

### 3.1. General Morphology of Antennae in Nymphs and Adults of H. patellifera

The antennae of all nine nymphal stages and adults were filiform (Figure 1 and Figure 2), consisting of three segments. The scape was located at the base of the antennae and longer than the pedicel. The Böhm’s bristle and a few sensilla chaetica were distributed on the surface of the scape and pedicel. The flagellum was the longest segment in the antennae, the width gradually narrowing from the base to the tip (Figure 2). With the growth of instars, the number of flagellomeres gradually increased. The flagellum consisted of about 44–110 flagellomeres from the 1st to the 9th instar, with about 100–120 in adults (Figure 2).

With the growth of instars, the elongation of antennae exhibited a regular pattern in development. The most obvious individual growth of antennae occurred from the 5th to 6th-instar nymphs, including an increase in the length of antennae and the width and length of the scape, pedicel, and first flagellomere. In male nymphs, the antennae length of the 6th-instar nymphs was approximately 1.25 times that of the 5th-instar nymphs. In contrast, it was 1.12 and 1.11 times in the 4th- to 5th-instar nymphs and 6th- to 7th-instar nymphs, respectively. The length and width of the scape and pedicel of the 6th-instar nymphs increased by approximately 1.3 and 1.2 times compared to the 5th-instar nymphs, respectively (Appendix A). The width and length of the first flagellomere reached about 1.1 and 1.2 times those of the 5th-instar nymphs.

The length of the male adult antennae was approximately 1.2 times that of the female. For nymphs, significant sexual dimorphism started to appear from the 7th instar, with the length of male antennae being longer than females (Figure 2). For females, there was no significant variation in the length of the antennae in adults after the 7th instar (Appendix A).

### 3.2. Types and Morphology of Sensilla in the Nymphs and Adults

Based on the morphology of external features observed with the SEM, overall, there were seven morphologically distinct types and four subtypes of sensilla in nymphs and adults of *H. patellifera.* Four subtypes of sensilla were identified in the nymphs (sensilla trichodea: StI, StII; sensilla campaniformia: ScaI, ScaII), and the undefined cuticular structure was only found on the tip of the 9th instar. The sensilla of adult antennae were almost consistent with that of nymphs, but the undefined cuticular structure was only observed in nymphs.

#### 3.2.1. Sensilla Chaetica (Sc)

Sensilla chaetica were the longest sensilla found in *H. patellifera.* Sc had a longitudinally grooved wall on the surface and a flexible socket at the base. Sc exhibited considerable diversity in length, including elongated, robust, and intermediate forms (Figure 3a–e). Sc presented on basal flagellomeres display robust forms (Figure 3a), while those occupied distal regions show elongated forms (Figure 3b). Intermediate-length Sc were distributed across distal flagellomeres, pedicel, and scape regions. (Figure 3b and Figure 4k).

#### 3.2.2. Sensilla Trichodea (St)

Sensilla trichoidea were the most numerous sensory structures on the antennae. They were similar to a slender hair and slightly curved into an arc shape. The base of each sensillum was inserted into a round concave socket, and their surfaces were multiporous (Figure 3h). Based on their length, location, and morphology, St were categorized into two subtypes. StI exhibited a sharper apex with pronounced curvature (Figure 3f). StI were mainly distributed along all margins of each flagellomere (Figure 4a), with an additional presence in the middle region of individual flagellomeres (Figure 4b). In contrast, StII possessed a blunter apex and demonstrated less pronounced curvature overall (Figure 3g). StII were located at the distal ends, proximal bases, and middle regions of each flagellomere (Figure 4a,j). In adults, the length of StI measured approximately 40–50 μm, while StII ranged from 27 to 30 μm (Appendix A).

#### 3.2.3. Sensilla Coelocapitula (Sco)

Sensilla coelocapitula were surrounded by a ring-shaped structure (Figure 4c). The ring was smooth, with a central depression, and within it grew a nipple-like protrusion. There was a small hole in the middle of the nipple-like protrusion (Figure 4d). They were mainly distributed on the distal end of each flagellomere of nymphs and adults (Figure 4b).

#### 3.2.4. Grooved Peg Sensilla (Sgp)

Grooved peg sensilla were short, with many longitudinal grooves, and were situated in a depression formed by the cuticle (Figure 4e). These sensilla terminal structures had an apical pore (Figure 4f). They were present in the distal part of the flagellomeres of nymphs and adults (Figure 4a).

#### 3.2.5. Böhm’s Bristle (Bb)

The shape of Böhm’s bristle was a short nail with a smooth surface, and it slantingly grew in the socket formed by the cuticle (Figure 4g). Böhm’s bristles were observed bilaterally on the scape and pedicel base of the nymph and adult antennae (Figure 4k).

#### 3.2.6. Sensilla Basiconica (Sb)

Sensilla basiconica were cone-shaped with a blunt tip. Each sensillum had a perforated surface and a non-flexible socket (Figure 4h,i). Sb were mainly distributed on the distal part of the lateral and anterior margins of each flagellum in nymphs and adults (Figure 4j).

#### 3.2.7. Sensilla Campaniformia (Sca)

The campaniform sensilla have a dome-like structure. Based on their external morphology, we divided the Sca into two subtypes. ScaI were surrounded by epidermal elevations and were flatter in appearance than ScaII (Figure 5a–c). They were present on the junction of two flagellomeres (Figure 5a) or closed to the base of flagellomeres in nymphs and adults (Figure 5b). Each ScaII had a hole in its middle part (Figure 5d), which was observed on the base of the antennal pedicel (Figure 5f).

#### 3.2.8. Undefined Cuticular Structure

Undefined cuticular structures were similar to a campaniformia sensilla but did not have the typical characteristics of a Sca. It might be a Sca that was gradually developing, or it could be an epidermal structure without sensory function (Figure 5e). These structures were only present on the dorsal surface of flagellomeres in the last-instar nymphs of *H. patellifera* (Figure 5g). Regarding its specific classification, further research is still needed in combination with the imaging results of the projection electron microscope.

### 3.3. The Distribution of Sensilla on the Antennae of Adults

On flagellomeres #1 to 14, Sc began to appear in both sexes. At the near of flagellomere #15 to 17, a large number of Sgp and some St started to appear on each flagellomere in males. In females, Sgp and St were present at the distal end of each flagellomere #27. At flagellomeres #37 to 45, St were the majority in females, and Sgp were dominant in males. The number of Sgp slightly reduced at the flagellomere #85 in males, and the sexual dimorphism of sensillar distribution disappeared in both sexes (Figure 6a–l).

### 3.4. The Distribution of Sensilla on the Antennae of Nymphs

Compared with the antennae of adult males, the Sgp and some St were present at the distal end of flagellomere #50 in the 9th-instar males (Figure 6g_1_). Additionally, the number of Sgp in the 9th-instar males was fewer than in adult males, and other types of sensilla displayed a similar pattern as adult males from flagellomere #58 to 60 (Figure 6i,i_1_). In 9th-instar females, the distribution of sensilla was similar to that in adult females, except the number of Sgp was slightly less than in adult females (Figure 6j,j_1_). Between the 9th-instar nymphs, there was no distinct sexual dimorphism in the distribution of sensilla. The distribution pattern of sensilla in other instars of nymphs was the same as in 9th-instar nymphs. There was no obvious sexual dimorphism between the males and females (Figure 6a_2_,e_2_,g_2_,i_2_,k_2_).

## 4. The Segmentation of the Flagellum

The sensillar distribution pattern varied along the longitudinal axis of the antennae and between the sexes. In order to represent these variations precisely, we proposed a new system to discriminate different parts of the flagellum. We divided the flagellum of adults into six parts based on the morphology of the flagellomere and the distributional features of the sensilla, from the most proximal (Part I) to the most distal (Part VI) (Figure 6). Each part was characterized as follows.

Part I (Flagellomeres #1~14)

Sc were distributed along a single row in the distal region of each flagellomere in both sexes. There were no other types of sensilla in this part and no obvious sexual dimorphism (Figure 6a,b).

Part II (Flagellomeres #14~17/18)

The first clear sexual differences were a large number of Sgp in this part in adult males (Figure 6c). In contrast to males, females had only Sc (Figure 6d).

Part III (Flagellomeres #19~27)

The flagellomere gradually increased in length between the sexes and then underwent a sudden decrease in this part. Based on this characteristic, the flagellomeres #19 to 27 were included in Part III. The length of flagellomeres in this part was reduced by about 50 μm in males (Figure 6e) and 100 μm in females (Figure 6f). The width of flagellomeres was reduced by about 10 μm in both sexes.

Part IV (Flagellomeres #28~37/45)

St were first present in this part in both sexes (Figure 6g). Sgp first occurred at the end of the flagellomere in females (Figure 6h).

Part V (Flagellomeres #37/45~75)

There were differences between the sexes in this part. In males, a large number of Sgp were dominant (Figure 6i), and St were generally distributed in females. The number of St and Sgp were obviously increased in females (Figure 6j).

Part VI (Flagellomeres #76~End)

Although the Sgp were dominant on the flagellomeres from Parts II to VI in males, the distribution of sensilla exhibited a similar pattern between the sexes in Part VI. The number of Sgp was only slightly decreased in males (Figure 6l).

Based on the distribution of sensilla in adults, the sensillar distribution of nymphs was also investigated. The main difference is the flagellomeres of male nymphs without abundant Sgp. Referring to the adult antennae, five different parts were distinguished in nymphs; the flagellar region of nymphs lacked Part II of adults. The distribution of sensilla in other instars in nymphs was the same as the 9th-instar nymphs and 3rd-instar nymphs. Therefore, we used the description of the flagellar region in the 9th-instar nymph and 3rd-instar nymph as representative of all other nymphal instars.

Similar to adults, Sc were distributed along a single circular line in the distal region of each flagellomere in the first part of the nymphs’ flagellum (Figure 6a_1_,b_1_).

In the second part of the nymphs’ flagellum, the length of flagellomeres increased gradually in nymphs but was suddenly reduced near the end of flagellomere #15. Based on this characteristic, we used this as a criterion to distinguish this part. Sc were only observed in nymphs and adult females (Figure 6e_1_,f,f_1_), and adult males began to exhibit St and Sgp in this part (Figure 6e).

In the third part of the nymphs’ flagellum, St and Sgp began to occur at the end of the flagellomeres in both male and female nymphs (Figure 6g_1_,h_1_). The distribution of sensilla was similar in female adults and nymphs. Adult males possessed more Sgp than male nymphs in Part III (Figure 6g,g_1_).

In the fourth part of the nymphs’ flagellum, the number of St and Sgp increased between nymphs (Figure 6i_1_,j_1_). The pattern of distribution of sensilla in adults was identical to that in nymphs (Figure 6i,j).

In the fifth part of the nymphs’ flagellum, the distribution of sensilla was identical to that in Part IV (Figure 6k_1_,l_1_). The length of sensilla was obviously increased in the nymphs of males and females, and the number of sensilla was basically stable in this part.

## 5. Widths and Lengths of Flagellomeres

### 5.1. Widths

The width of flagellomeres was gradually reduced from the base to the tip of the flagellum in a regular manner (Figure 7a,b). Until the 7th instar, the width of the flagellomeres had an obvious distinction between the sexes (Figure 7a). In contrast, after the 6th instar and adults, males evidently bore wider flagellomeres than females, except the terminal part of the antennae (Figure 7a,b).

The width of the first flagellomere gradually increased progressively with each successive molt. The variation in width in the first flagellomere among the 1st to 3rd instars was small. There was obvious variation in width until the 6th instar. The appearance of abdominal sexual difference at the 4th instar was followed by an enlargement of the first flagellomere in males compared to females and more visibly at the 7th instar. However, the width of the first flagellomere was less in adults than in the 9th instar, and the males were more remarkable than females in the degree of reduction (Figure 7c).

We investigated the differences in the width of flagellomeres at the distal parts of antennae from 1st instar to adult by averaging the width between the 10th and 30th flagellomeres via beginning to calculate from the antennae tip. The width was not significantly different from the 1st to 5th instar. However, females and males displayed an enlargement of the distal part of the antennae at the transition between the 5th and 6th instars. In addition, the width of flagellomeres had obvious differences between the sexes at the 7th instar. There were also significant differences in the width of flagellomeres of adult males and females (Figure 7d).

### 5.2. Lengths

In adults, the antennae displayed a noticeable distinction in length between males and females. The corresponding flagellomeres in males were significantly longer than in females, except for the proximal parts of the antennae (Figure 8a,b). There was no uniformity in the curves of the length of flagellomeres, but the model of the variation was quite similar between males and females: a gradual increase from flagellomeres #1 to #14 reaching an asymptote, a second increase until a peak around flagellomere #19/27, and then a sudden decrease (valley) in both sexes. After the sudden reduction, the length gradually increased until the antennal tip segment in females, while it increased to a larger degree in males before a gradual diminution from about flagellomere #70 (Figure 8a). The length of flagellomeres in Part I reached an asymptote (Figure 8b). We could unambiguously identify the transition point between Parts I and II. The model of variation of the length of flagellomeres was similar in nymphs (Figure 8c,d).

The length of the first flagellomere was constant from 1st to 3rd instars. The extension of the first flagellomere was observed from the 4th instar, and notable differences began to occur at the 6th instar. In addition, the length of the first flagellomere had obvious differences at the 7th instar between the sexes, which also appeared as obvious differences in adults in both sexes (Figure 8e).

The length of the flagellomeres gradually increased in successive instar. There was no obvious sexual dimorphism from the 4th to the 6th instar, until the 7th instar, when there were significant differences in the length of the flagellomeres between the sexes (Figure 8f).

## 6. Addition of Flagellomere

The addition of flagellomere had two obvious sites in nymphs and adults of *H. patellifera.* The first partial division started at the end of the first flagellomere in every instar and in the adult (Figure 9a). The first flagellomere was elongated and displayed a distinct segmentation line, while the following segments were short, and the length gradually increased (Figure 9b). The second partial division appeared at or near 10~20 flagellomeres (between Part II and Part III) in nymphs and adults (Figure 8a,c and Figure 9c). We observed obvious cleavage, and the length of the flagellomere was suddenly reduced at this point (Figure 8b,d). Based on this mechanism, we created a schematic example of antennal development. Figure 10 shows the process of increasing the number of flagellomeres in these two sections. In this example, the antennae would develop as follows. The number of the following segments in the first flagellomere increased in an irregular manner (Figure 10a). An increase of 2 flagellomeres of Part IV would mean that 2 flagellomeres from Part III developed olfactory sensilla and become flagellomeres of Part IV (Figure 10b). Thus, to keep Part III constant, a single flagellomere from Part II divided into two flagellomeres, henceforth belonging to Part III, which compensated for the two flagellomeres that matured (Figure 9d,e and Figure 10b). Finally, to keep Part I constant, the meriston would produce a single flagellomere in addition to a new meriston (Figure 10a,b). The segmentation of these flagellomeres increased their length at these two locations (Figure 9a–d).

## 7. Discussion

### 7.1. The Diversity of Antennal Sensilla

There is significant variation in the diversity of sensilla present in different insect taxa [20]. For example, Lepidoptera (moths and butterflies) have documented up to 9–11 distinct sensilla types [21,22], while some parasitic insects have around 3–5 sensilla types [23,24]. The types of sensilla found in *H. patellifera* are very similar to other mantid species; there are around seven types of sensilla [11,13]. This indicates that the diversity of sensilla types in mantids occupies an intermediate position among insects. Previous studies reported that the diversity of antennal sensilla may be a result of environmental pressures leading insects to adapt to their habitats [25].

In most insects, Sc exhibit diversity in length and morphology [26]. In this study, Sc demonstrated morphological homogeneity while exhibiting considerable diversity in length, including elongated, robust, and intermediate forms. For example, in adults, elongate Sc measured 105–125 μm, intermediate forms 60–80 μm, and robust forms 28–55 μm (Appendix A). This result is consistent with those reported in *T. aridifolia*, where analogous Sc forms were documented: elongate (~110 μm), intermediate (60–70 μm), and robust (25–50 μm) [11]. There are no significant differences among nymphs and adults in the antennae and antennal sensilla of *H. patellifera*. However, the density of these sensilla increases with instars. The greatest density of sensilla occurs along the length of the flagellum, with the greatest densities typically found toward the distal end and on the lateral extensions to the flagella [27]. The density of St and Sgp are most abundant in adult mantids. Both types of sensilla occupy the highest density at the distal antennae [11,13]. St and Sgp may be able to detect female-emitted sex pheromones more efficiently, as shown previously [12]. In addition, an undefined cuticular structure was observed in the last nymph. The epidermal structure and functions in insects are diverse. Innervated cuticular structures have sensory roles in detecting mechanical stimuli, chemical stimuli, electric fields, temperature, and humidity [28]. As for nonsensory structures, they can confer photonic, thermoregulatory, hydrophobic, and hydrophilic properties to the cuticle [29]. Whether this type of sensilla has sensory functions needs to be further studied in combination with transmission electron microscopy.

### 7.2. On Sexual Dimorphism and Antennal Development in H. patellifera

In hemimetabolous insects, the insects’ developmental pattern goes through gradual morphological changes, and the nymphal stages morphologically resemble adults, while sexual differentiation, in general, appears in later instars [30]. Mantids are hemimetabolous insects. The research on sexual differentiation of mantids mainly focuses on antennae. Carle et al. first revealed that the antennal dimorphism of *T. aridifolia* (Mantodea: Mantidae) significantly occurs at the 6th instar [14]. In contrast, significant sexual dimorphism is visible from the 7th instar in *H. patellifera* (Figure 7a–d and Figure 8e,f). Considering the interspecific variation of nymph instar number, the results are consistent; the significant sexual dimorphism of mantid antennae begins to occur in last-instar nymphs. Interestingly, the simultaneous appearance of antennal development and abdominal segmentation in *H. patellifera* suggests that there may be a hormone in insects that induces this phenomenon. The ecdysone is directly implied in antennal development and especially in molting in insects [31]. Moreover, its role plays an essential role in establishing sexually dimorphic in insects. For example, in the Japanese mealybug *Planococcus kraunhiae* (Kuwana), ecdysone regulates the development of males into winged adults, while females emerge as neotenic wingless adults [32]. This phenomenon might provide interesting points for future research studies of mantids.

In insects, the sexual dimorphism of antennae displays many forms, such as the shape and size of the antennae and the diversity, abundance, and distribution of sensilla on the antennae [27]. The most direct factor that distinguishes the sexes in mantids lies in the longer antennae in males [12,14]. This may be because males have more flagellomeres than females [33]. However, there are few studies on the measurement of the length of individual flagellomeres in insects. In this study, we examined the differences in length between the sexes of each corresponding flagellomere in adults. These differences are not uniform along the antennae. The greatest differences occur in the middle section (Parts II to VI), where a large number of grooved peg sensilla are located [11]. Thus, sexually dimorphic antennae can also be distinguished by the number and distribution of types of sensilla [34]. The phenomena described above are similar in several Mantidae species [12,14].

The antennal development of termites and cockroaches has been studied in recent years. In the antennae of termites and cockroaches, the primary flagellomeres divide once, producing two secondary flagellomeres each [35,36]. In contrast, in the antennae of *H. patellifera*, the number of newly formed flagellomeres from the first flagellomere is irregular, but the manner in which flagellomeres are added at this position is the same. However, in grasshoppers and mantophasmids, the first flagellomere consistently generates a new meristal segment with each molt [37,38]. Compared to termites and cockroaches, the *H. patellifera* antennae have a second site to add the flagellomeres. A single flagellomere from the secondary partial site regularly divides into two flagellomeres, thereby increasing the number of distal partial flagellomeres. Although a secondary site for antennae growth was also previously discovered in grasshoppers during their early development, this pattern is different from the mantids [38,39]. In grasshoppers, the secondary site divides into multiple flagellomeres at a given molt [40]. Antennal development patterns further suggest that mantids are closely related to termites and cockroaches.

### 7.3. Comparison of Antennae Features with Other Insects

Details of the types of sensilla and their spatial distribution have been described for some mantids [11,13,14], focusing on the species from Hymenopodidae and Mantidae. Comparing the adult antennae of *H. patellifera* and *T. aridifolia*, no significant differences were observed in the external morphology except for the length of the male antennae. The average length of antennae in *T. aridifolia* is longer than that of *H. patellifera* [14]. However, comparing the antennae of *H. patellifera* and *C. nebulosa,* the external morphology of the antennae shows significant differences. All flagellomeres in males of *H. patellifera* are cylindrical with equal width, while the flagellomeres of *C. nebulosa* are nearly cone-shaped, contracting proximally and widening distally [13]. Additionally, in adult *H. patellifera*, the length of the antennae in adult males is about 1.2 times that of females (Appendix A), while *C. nebulosa* males have antennae 1.5 times the length of females. Therefore, the sexual dimorphism in the length of antennae in *C. nebulosa* is more remarkable than in *H. patellifera.*

The diversity of sensilla types on the flagellum often leads to imprecise localization of each sensillum. In order to represent these variations precisely, Carle et al. developed a nomenclature to discriminate different parts of the flagellum [11]. The flagellum of *C. nebulosa* adults is divided into five parts [13], while *H. patellifera* and *T. aridifolia* are divided into six parts. The major distinction between the two antennae types is the different distribution of Sgp. In the males of *H. patellifera* and *T. aridifolia,* the Sgp occurs in Part II of the flagellum, while the Sgp begins to appear in Part I of *C. nebulosa* antennae flagellum. *C. nebulosa* prefers tropical rainforests with a warm and humid climate, while *H. patellifera* and *T. aridifolia* are common species in the Palaearctic region. We speculate that the difference in the number and distribution of Sgp may reflect the distinct living environments of *H. patellifera*, *T. aridifolia*, and *C. nebulosa*. *H. patellifera* and *T. aridifolia* belong to different subfamilies but have similar antennae morphological characteristics, distinct from *C. nebulosa* (Hymenopodidae). It is indicated that the antennae external morphology, types, and distribution of sensilla of antennal sensilla could be diagnostic features among the families of Mantodea. Furthermore, the density of Sgp is significantly higher in males compared to females in *H. patellifera*. These characteristics seem common to other mantid species, such as *P. albofimbriata* and *T. aridifolia* [11,12]. Research has shown that males with a higher number of Sgp are able to detect pheromones more rapidly, which enhances their chances of successful mating [12,14].

Cockroaches (Dictyoptera: Blattodea) are the closest known relatives to mantids [41]. Research on the nymph’s antennae of these organisms has shown that their antennae are filiform. Moreover, no sex-related differences were observed in the distribution pattern of sensilla in nymphs [42]. These results are similar to those observed in the antennae of *H. patellifera* nymphs. However, the sensilla type in adults between mantids and cockroaches has some differences. Antennae of adult male cockroaches, sw-A and sw-B sensilla, are two subtypes of perforated sensilla basiconica, which have been identified as olfactory sensilla [43]. Among these, the sw-B sensillum is also sex pheromone-receptive [42]. The numbers and distribution patterns of sw-A and sw-B sensilla drastically change after the final molt. In males, sw-B sensilla drastically increase throughout all of the antennal flagellomeres after the final molt, whereas females have a greater number of sw-A sensilla [42]. In mantids, the Sgp are considered a type of sex pheromone receptor sensilla [11]. In our study, during the last molting, the large number of Sgp is dominant in adult males, and St are generally distributed in adult females. The sensilla types of sex pheromone reception are different between mantids and cockroaches despite their close relationship. This may be related to their different lifestyles. Praying mantids are solitary predators that typically survive on a carnivorous diet [44]. In contrast, most cockroaches are gregarious, consume an omnivorous diet [45], and are usually found in tropical or other mild climates [46]. We speculate that in order to adapt to this living environment, they may have evolved sensilla that are most beneficial to themselves.

## 8. Conclusions

In this study, we conducted a detailed observation of the antennal sensilla types and distributional patterns across different developmental stages of *H. patellifera* using SEM. We identified seven types and four subtypes of sensilla on the antennae of *H. patellifera*. Our results reveal significant sexual dimorphism in mantid antennae. This dimorphism begins to emerge in elderly nymphs and becomes pronounced in adults, with males exhibiting larger flagellomeres and a greater abundance of grooved peg sensilla. Furthermore, we identified two key growth sites critical for antennal development in mantids. This research not only enhances our understanding of antennal morphology and development in Mantodea but also contributes to the understanding of mantid taxonomy and evolution.

## Figures and Tables

**Figure 1 insects-16-00655-f001:**
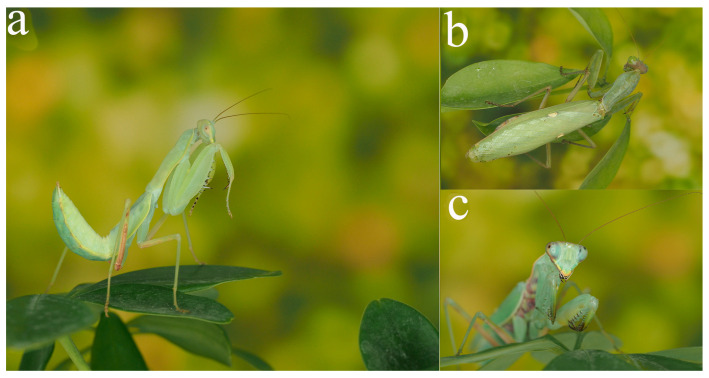
Nymph and adult habitus of *H. patellifera*. (**a**) 6th instar male nymph in lateral view; (**b**) female adult in dorsal view; (**c**) head of male adult in anterior view.

**Figure 2 insects-16-00655-f002:**
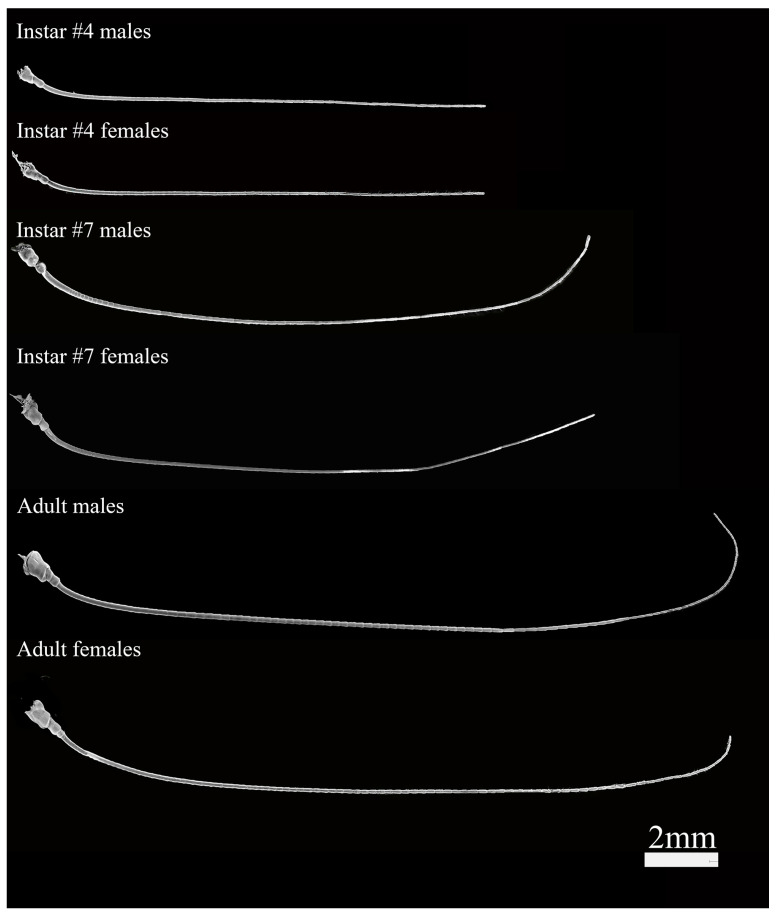
Antennae of *H. patellifera* nymphs and adults in ventral view. (including 4th and 7th instars and adult antennae).

**Figure 3 insects-16-00655-f003:**
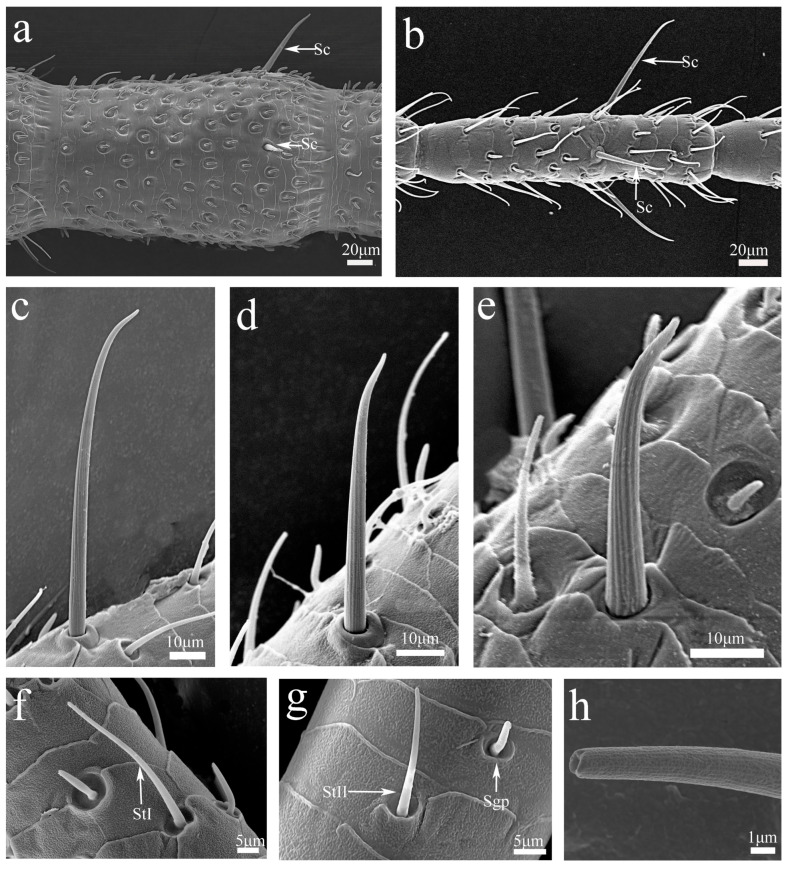
SEM micrographs of antennal sensilla and flagellomeres of *H. patellifera*, 6th-instar nymph male and adult. (**a**) The proximal flagellomere with sensilla chaetica (Sc), adult male antennae; (**b**) the apical flagellomere with Sc, 6th female antennae; (**c**) slender Sc located near the 80th flagellomere, in lateral view; (**d**) medium-length Sc located near the 50th flagellomere, in lateral view; (**e**) robust Sc located near the 30th flagellomere, in lateral view; (**f**) sensilla trichodea I (StI); (**g**) sensilla trichodea II (StII); (**h**). enlarged view of St with a multiporous aspect.

**Figure 4 insects-16-00655-f004:**
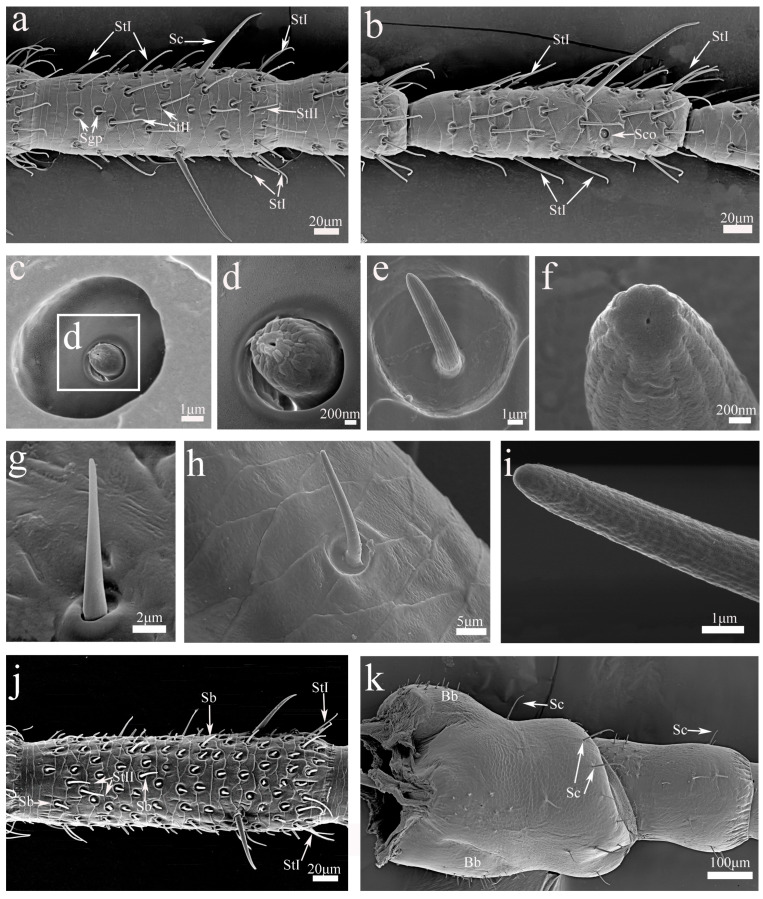
SEM micrographs of antennal sensilla, flagellomeres, scape, and pedicel in nymphs and adults of *H. patellifera*. (**a**) Flagellomere near the base of the antennae from an 8th-instar male nymph; (**b**) flagellomere near the distal part of antennae from an 8th-instar male nymph; (**c**) sensilla coelocapitula (Sco); (**d**) enlarged view of the mid-section of Sco; (**e**) grooved peg sensilla (Sgp); (**f**) enlarged view of the Sgp with an apical pore; (**g**) Böhm’s bristle (Bb); (**h**) sensilla basiconica (Sb); (**i**) enlarged view of the multiporous aspect of Sb; (**j**) distal part of a flagellomere from male antennae; (**k**) scape and pedicel of the right antennae of an adult male in ventral view.

**Figure 5 insects-16-00655-f005:**
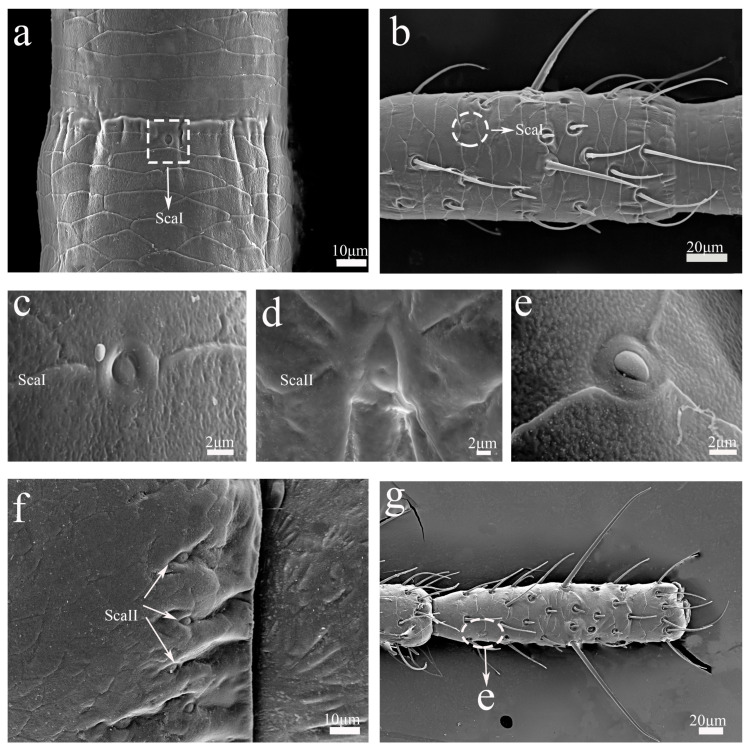
SEM micrographs of antennal sensilla, flagellomeres, and the pedicel in nymphs and adults of *H. patellifera*. (**a**) campaniformia I (ScaI), antennal flagellomeres of the base part of adult male in ventral view; (**b**) campaniformia I (ScaI), antennal flagellomere of 6th-instar female nymph in ventral view; (**c**) sensilla campaniformia I (ScaI); (**d**) sensilla campaniformia II (ScaII); (**e**) undefined cuticular structure; (**f**) campaniformia II (ScaII), antennal pedicel of adult male in ventral view; (**g**) the terminal flagellomere, 9th-instar female nymph antennae.

**Figure 6 insects-16-00655-f006:**
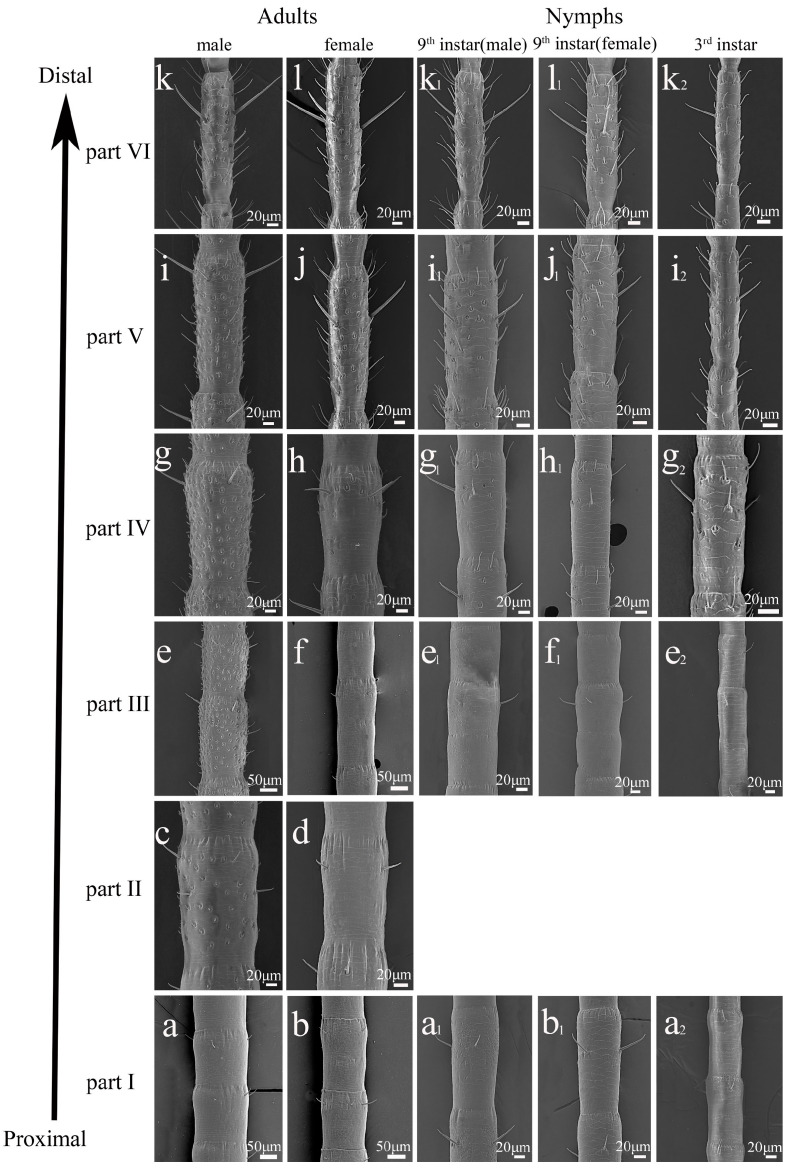
Distribution of sensilla along the antennae in adults of *H. patellifera.* Based on the distribution of sensilla along the antennae, six different parts were distinguished (**a**–**l**). Based on the adult antennae, five different parts were distinguished in nymphs (from (**a_1_**) to (**l_1_**)). Abbreviation: Micrographs of the different parts at the 9th instar (**a_1_**,**b_1_**,**e_1_**–**l_1_**) and 3rd instar (**a_2_**,**e_2_**,**g_2_**,**i_2_**,**k_2_**). Notes: SEM images (**a**,**c**,**e**,**g**,**i**,**k**) represent six parts of adult males, while SEM images (**b**,**d**,**f**,**h**,**j**,**l**) represent six parts of adult females. SEM images (**a_1_**,**e_1_**,**g_1_**,**i_1_**,**k_1_**) represent five parts of 9th-instar males, while SEM images (**b_1_**,**f_1_**,**h_1_**,**j_1_**,**l_1_**) represent five parts of 9th-instar females.

**Figure 7 insects-16-00655-f007:**
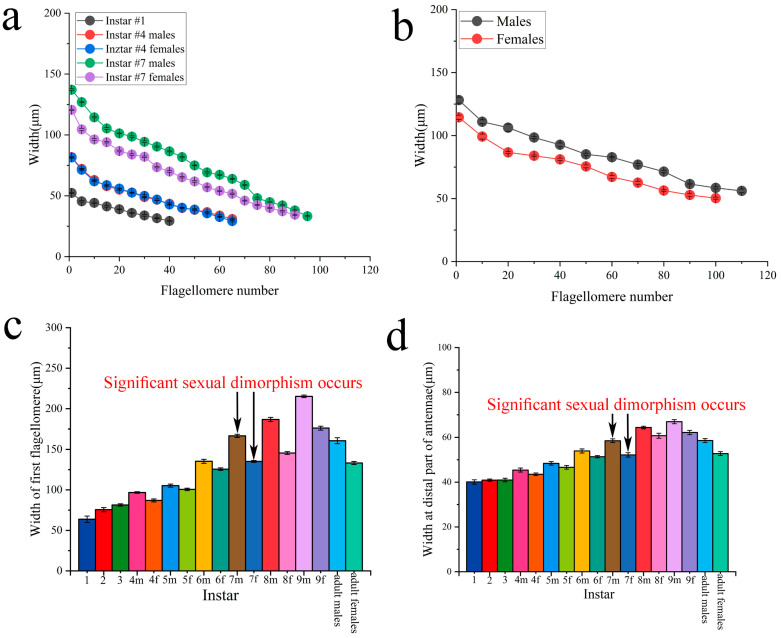
Width of flagellomeres in nymphs and adults of *H. patellifera*. The curves show the width of flagellomeres measured every 5 segments at the 1st, 4th, and 7th instars (**a**), and every 10 segments in adults (**b**); (**c**) width of the first flagellomere for nymphs and adults; (**d**) width at the distal part of antennae (average of flagellomeres placed between the 10th and 30th flagellomeres from the distal end) for nymphs and adults. Notes: The number represents instar in (**c**,**d**), m and f indicate gender. m indicates males, and f indicates females.

**Figure 8 insects-16-00655-f008:**
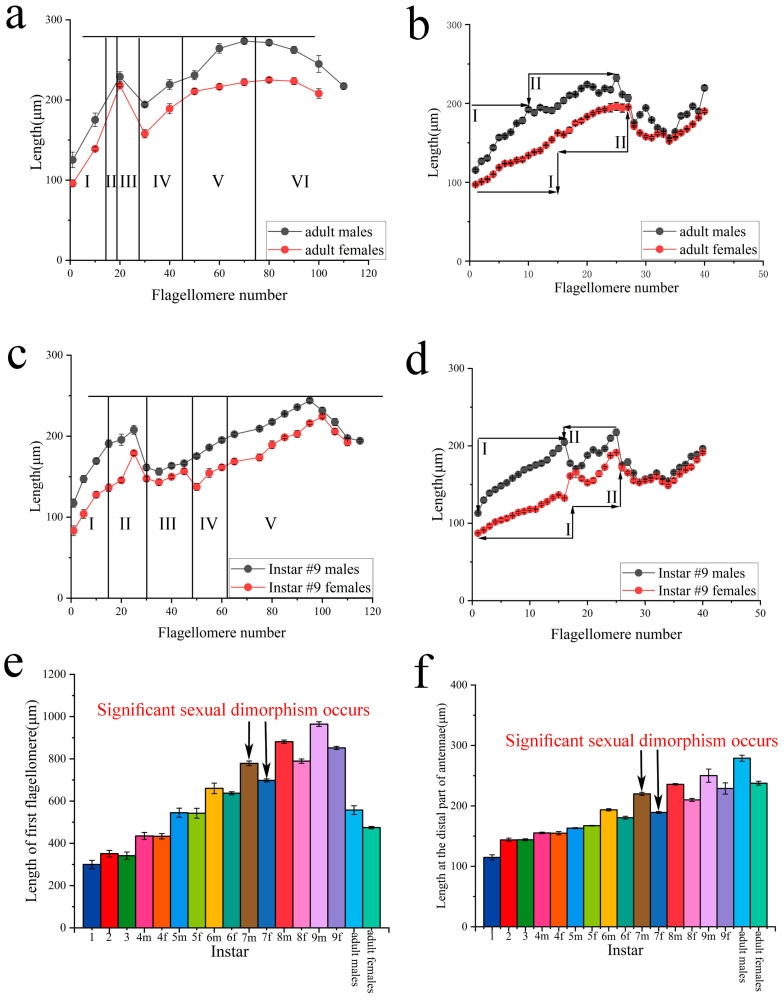
Length of flagellomeres in nymphs and adults of *H. patellifera*. (**a**,**b**) The curves show the length of flagellomeres measured every 10 segments (**a**) and every segment (**b**) from the proximal part of antennae in adult males and females; (**c**,**d**) the curves show the length of flagellomeres measured every 5 segments (**c**) and every segment (**d**) from the proximal part of antennae in the 9th-instar nymph between sexes; (the length of flagellomeres of other instars are similar, only data for antennae of 9th instar antennae are shown here (**e**) length of the first flagellomere for nymphs and adults; (**f**) length at the distal part of antennae (average of flagellomeres placed between the 10th and 30th flagellomeres from the distal end) for nymphs and adults. Notes: In (**e**,**f**), the number represents nymph instar; m and f indicate gender. m indicates males, and f indicates females.

**Figure 9 insects-16-00655-f009:**
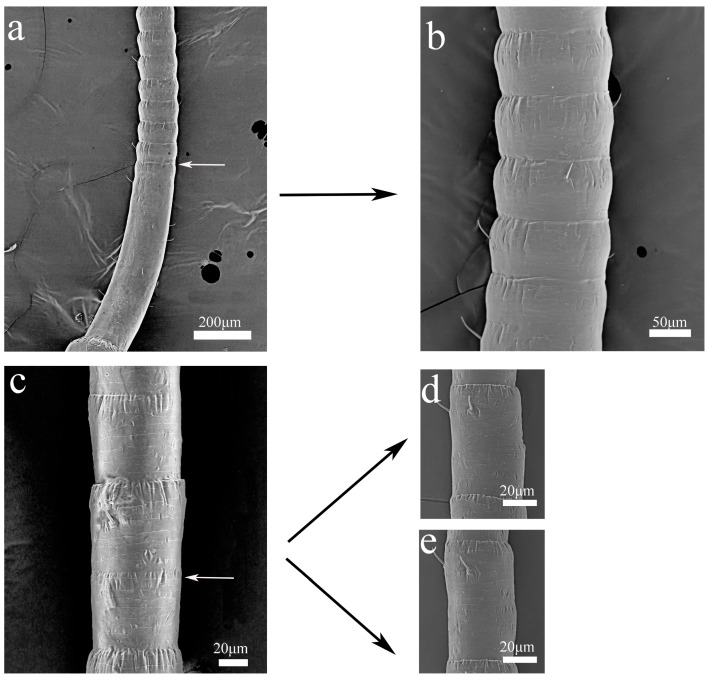
SEM images represent the sections of two growth sites derived from *H. patellifera*. (**a**) The first site occurs at the end of the first flagellomere; (**b**) at the end of the first flagellomere by the division of the first flagellomere, which presents many rows of Sc as shown on the SEM micrographs; (**c**–**e**) the second site of genesis is located at the transition between Parts II and III, where a single flagellomere of Part II (**c**) divides into two flagellomeres (**d**,**e**) of Part III after molting. Notes: The white arrow indicates the point at which segmentation commenced, while the black arrow denotes the flagellated subsegment that was formed subsequent to the segmentation process.

**Figure 10 insects-16-00655-f010:**
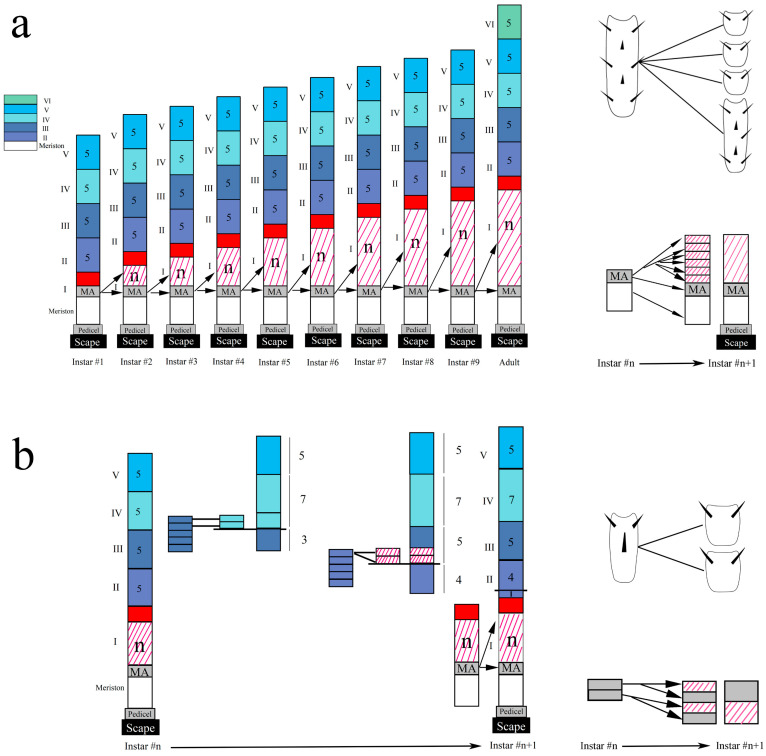
Mechanism of addition of flagellomere. This model, using the number 5 to represent the number of flagellomeres in each part from Parts II to VI, details the process of antennal development in these two sections. (**a**) This diagram shows a part of flagellomeres, newly formed during molting, is generated at the proximal end of antennae by the division of the meriston. Red squares represent the initial number of flagellomeres in Part I, and n represents an uncertain number of flagellomeres produced by MA splitting. Red lines on white background represent an increase in the total number of flagellomeres; (**b**) looking at the features of flagellomeres along the antennae, another location of genesis lies at the transition between Parts II and III, where a single flagellomere of Part II divides to give two flagellomeres of the Part III after molting. Notes: Meriston: the first flagellomere; MA: meristal annulus.

## Data Availability

The authors confirm that the data supporting the findings of this study are available within the article and its Appendix A.

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
