# Peer review of "Antennal Sensilla Morphology and Flagellomere Addition in Nymphs and Adults of *Hierodula patellifera* Serville, 1839 (Mantodea: Mantidae)"

_insects, 2025, doi:10.3390/insects16070655_

Round 1

Reviewer 1 Report (New Reviewer)

Comments and Suggestions for Authors

General Comments

The original work of Huan Wang et al .a thorough study using scanning electron microscopy to describe the antennal sensilla types and distributional patterns across different developmental stages in H. Patellifera. The investigation results shows the morphology of the antennae are filiform consisting of three segments which are Scape,Pedicel and Flagellum. The flagellum contains 44 to 120 flagellomeres, increasing with growth stages and also with instars. They also identified seven types and ten subtypes of Sensilla including sensilla trichodea (StI, StII), sensilla basiconica (Sb), sensilla chaetica (Sc), sensilla coelocapitula (Sco), sensilla campaniformea (ScaI, ScaII, ScaIII), sensilla grooved peg (Sgp), and Böhm bristle (Bb). ​ ScaIII is exclusive to nymphs. Significant sexual dimorphism in antennae begins at the 7th instar, with males having longer antennae, larger flagellomeres, and more sensilla grooved peg (Sgp) and ecological adaptation have been identified. These findings contribute to understanding mantid taxonomy and evolution. In conclusion, į believe that the article would significantly contribute to the field. However, į have a few points that į would like to address, and į expect the authors to incorporate them in the final version. Addressing these points will enhance the article’s clarity and presentation for the readers.

Comments:

1. Line 34: The abstract is concise and effectively summarizes the key findings. However, the statement “Notable sexual dimorphism begins to appear in antennae from the 7th instar“ this statement can be more precise and clear with the refers to its length, sensilla distribution, or both. For example “Sexual dimorphism in antennal length and sensilla distribution becomes evident from the 7th instar“.

2. Line 36: “Mantid antennae develop by the addition of new segments “ this statement can be specify that these are flagellomeres and clarify the sections for example., proximal or secondary sites.

3. Line 121-149: General morphology of antennae is well described, but the text is dense with measurements. All this date can be consider summarizing these in to a table improve readability.

4. Table S1 and S2 are referenced but not provided in the manuscript. Please include these supplementary tables or summarize key data in the main text.

5. Line 144: The study notes significant sexual dimorphism from the 7th instar. Are there any hypotheses or experimental evidence to explain why this dimorphism appears at this specific stage rather than earlier or later?

6. Line 196: how do the observed differences in sensilla grooved peg (Sgp) distribution correlate with functional roles in pheromone detection or other behaviors?

7. Some SEM images lacked sufficient scale bars or descriptions in the figure legends.

8. Figure 10 needs some more detailed schematic diagrams to explain the two growth sites of flagellomeres more clearly.

9. Not all abbreviations were clearly introduced at first mention. Example Bb, StI.

10.There are typographical and grammatical errors throughout the manuscript. Need a thorough proofreading would help improve the overall readability. For example “the the “ in simple summary

Over all, this study offers an important contribution to understand the highlights the morphological diversity of mantid antennae, their developmental patterns, and the role of sensilla in sexual dimorphism and ecological adaptation. ​ Addressing the points mentioned above would enhance the clarity and impact of the manuscript. I look forward to seeing the revised version.

Comments on the Quality of English Language

There are typographical and grammatical errors throughout the manuscript. Need a thorough proofreading would help improve the overall readability. For example “the the “ in simple summary

Author Response

Reviewer #1:

The original work of Huan Wang et al .a thorough study using scanning electron microscopy to describe the antennal sensilla types and distributional patterns across different developmental stages in H. Patellifera. The investigation results shows the morphology of the antennae are filiform consisting of three segments which are Scape, Pedicel and Flagellum. The flagellum contains 44 to 120 flagellomeres, increasing with growth stages and also with instars. They also identified seven types and ten subtypes of Sensilla including sensilla trichodea (StI, StII), sensilla basiconica (Sb), sensilla chaetica (Sc), sensilla coelocapitula (Sco), sensilla campaniformea (ScaI, ScaII, ScaIII), sensilla grooved peg (Sgp), and Böhm bristle (Bb). ScaIII is exclusive to nymphs. Significant sexual dimorphism in antennae begins at the 7th instar, with males having longer antennae, larger flagellomeres, and more sensilla grooved peg (Sgp) and ecological adaptation have been identified. These findings contribute to understanding mantid taxonomy and evolution. In conclusion, į believe that the article would significantly contribute to the field. However, į have a few points that į would like to address, and į expect the authors to incorporate them in the final version. Addressing these points will enhance the article’s clarity and presentation for the readers.

We appreciate your time in reviewing our manuscript. Detailed responses are provided below, along with the revised and corrected files.

Comment 1: Line 34: The abstract is concise and effectively summarizes the key findings. However, the statement “Notable sexual dimorphism begins to appear in antennae from the 7th instar“ this statement can be more precise and clear with the refers to its length, sensilla distribution, or both. For example “Sexual dimorphism in antennal length and sensilla distribution becomes evident from the 7th instar”.

Response: Thank you for your suggestion. We have changed the sentences as your suggestion in the abstract: “sexual dimorphism in antennal length, width and sensilla distribution becomes evident from the 7th instar”.

Comment 2: Line 36: “Mantid antennae develop by the addition of new segments “ this statement can be specify that these are flagellomeres and clarify the sections for example., proximal or secondary sites.

Response: We agree with this comment. We have changed the sentences “Mantid antennae develop by the addition of new segments” in the manuscript to “Mantid antennae develop by the addition of new flagellomeres at their proximal and secondary division sections”.

Comment 3: Line 121-149 : General morphology of antennae is well described, but the text is dense with measurements. All this date can be consider summarizing these in to a table improve readability. Table S1 and S2 are referenced but not provided in the manuscript. Please include these supplementary tables or summarize key data in the main text.

Response: Based on the suggestions of other reviewers, in order to simplify the content of the article, some data in this paragraph are presented in the form of bar charts in the article (Figure 7 and 8). Some other data are presented in the form of appendices ( Table S1), and both Appendices S1 and S2 are added to the end of the manuscript.

Comment 4: Line 144: The study notes significant sexual dimorphism from the 7th instar. Are there any hypotheses or experimental evidence to explain why this dimorphism appears at this specific stage rather than earlier or later?

Response: In hemimetabolous insects, the insects’ developmental pattern goes through gradual morphological changes and the nymphal instars are morphologicall similar to the adults, while sexual differentiation, in general, appears in the latest nymphal instars. For instance: (1) the research on sexual differentiation of mantids mainly focuses on antennae. Carle et al. first revealed that antennal dimorphism of T. aridifolia (Mantodea: Mantidae) significant occurs at the 6th instar [1]. the results is consistent in H. patellifera., the significant sexual dimorphism of mantid antennae began to occur on elderly nymph, and continue to strengthen. (2) The proportion of sex-biased genes gradually increases during development in T. californicum. Experiments show that there is usually a significant sex bias phenomenon in the latest nymphal instars [2]. (3) Furthermore, during the development of cockroaches, it was found that their body length showed a distinct sex dimorphism phenomenon in the latest nymphal instars. Meanwhile, in the final instar, the pair of styli upon coxosternum IX disappears. However, the males undergo no marked changes in the exposed parts of the postabdomen, neither noticeable reduction/extension of coxosterna nor loss of styli [3]. These results indicate that in hemimetabolous insects, sexual dimorphism usually occurs in the latest nymphal instars.

[1] Carle, T.; Yamawaki, Y.; Watanabe, H.; Yokohari, F. Antennal development in the praying mantis (Tenodera aridifolia) highlights multitudinous processes in hemimetabolous insect species. PLOS ONE 2014, 9, e98324.

[2] Dynamics of sex-biased gene expression during development in the stick insect Timema californicum. Heredity 2022, 129, 113-122.

[3] Reproductive biology and postembryonic development of a polyphagid cockroach Eucorydia yasumatsui Asahina (Blattodea: Polyphagidae). Arthropod Systematics and Phylogeny 2014, 72, 193-211.

Comment 5: Line 196: how do the observed differences in sensilla grooved peg (Sgp) distribution correlate with functional roles in pheromone detection or other behaviors?

Response: The groove peg sensilla shows obvious sexual dimorphism in adults. Male adults have more of this sensilla than females and appear earlier on the flagellum. Furthermore, it has been proved in previous literature on mantids that this type of sensilla is related to the detection pheromone. These results are described in the discussion.

Comment 6: Some SEM images lacked sufficient scale bars or descriptions in the figure legends.

Response:  The SEM images lacking the scale have been added with the corresponding scale, and the descriptions lacking in the sensilla part have also been supplemented accordingly.

Comment 7: Figure 10 needs some more detailed schematic diagrams to explain the two growth sites of flagellomeres more clearly.

Response: Thank you for your proposal. We have supplemented Figure 10 to have a clearer understanding of these two growth sites.

Comment 8: Not all abbreviations were clearly introduced at first mention. Example Bb, StI.

Response: Thank you for your suggestion. The missing part of the sensilla description has been supplemented.

Comments on the Quality of English Language

There are typographical and grammatical errors throughout the manuscript. Need a thorough proofreading would help improve the overall readability. For example “the” in simple summary.

Response: We have corrected typographical and grammatical errors throughout the article to improve its accuracy and readability.

Reviewer 2 Report (New Reviewer)

Comments and Suggestions for Authors

Dear authors,
The article presents significant data on the type and distribution of sensilla in the mantid species Hierodula patellifera, which is typical of morphological analyses in this context. Furthermore, the authors focused on the morphology and development of the antennae and proposed a new division system based on the number of flagellomeres and the distribution of the sensilla. The paper is well documented, with SEM images and statistical graphs. However, the figures are not cited in the correct order. I have some reservations about the classification of sensilla campaniformia, which are noted on line 210.
Some comments on the text are provided below:

 Line 1: This paper does not include any study of ultrastructures. The title should be: Morphology or ultramorphology
 Line 28: The number of subtypes has to be changed.
 Line 30: Such a distinction is very doubtful; comment in the paragraph on the campaniform sensilla.
 Line 124: What is a Bohm bristle- cuticular structure without sensory function, or are they sensilla?. It should be explained because in the description part of the sensilla, Bohm bristles are not described.
 Line 126: Fig. 7c doesn't confirm this description
 Line 154: What is a type sensilla in contrast to subtypes? According to this division, there are five subtypes (two sensilla trichodea and three campaniformea).
Line 160: How did the authors recognise the terminal pore, since it is not marked or visible in the photos?
Lines 162-163: Should be subtypes (marked also on the figures)
Line 209: What morphological features may indicate that these are sensory structures?
 Line 210: This character denies that such a structure should be classified as a campaniform sensillum (Sca III).
It may be just a cuticular structure without a sensory function.
 Line 211: On which flagellomeres these structures are found, there is no information in the distribution part of the sensilla.
Line 411: Authors should use  "subtypes".

Comments on the Quality of English Language

Many of the entries in the Results and Discussion sections are very long. There are problems with the use of the singular and plural forms, and it is often unclear whether one or many flagellomeres are being referred to.

Author Response

Reviewer #2:

The article presents significant data on the type and distribution of sensilla in the mantid species Hierodula patellifera, which is typical of morphological analyses in this context. Furthermore, the authors focused on the morphology and development of the antennae and proposed a new division system based on the number of flagellomeres and the distribution of the sensilla. The paper is well documented, with SEM images and statistical graphs. However, the figures are not cited in the correct order. I have some reservations about the classification of sensilla campaniformia, which are noted on line 210.

Response: Thank you for your detailed revisions of our manuscript. The content has been modified based on the feedback received.

Comment 1: This paper does not include any study of ultrastructures. The title should be: Morphology or ultramorphology.

Response:  We agree with this comment. We have changed the "ultrastructure" in the title to “Morphology”.

Comment 2: The number of subtypes has to be changed.

Response: We agree with this comment. The number of subtypes of the sensilla has been changed.

Comment 3: Line 30: Such a distinction is very doubtful; comment in the paragraph on the campaniform sensilla. What morphological features may indicate that these are sensory structures? Line 210: This character denies that such a structure should be classified as a campaniform sensillum (ScaIII). It may be just a cuticular structure without a sensory function. Line 211: On which flagellomeres these structures are found, there is no information in the distribution part of the sensilla.

Response: Thank you for your suggestion. The sensilla has been redescribed in the text. This sensilla is an uncertain cuticular structure, similar to campaniformia sensilla, but does not have the typical characteristics of Sca. It could be a gradually developing Sca or an epidermal structure without sensory function. These structures are only present on the dorsal surface of flagellomeres in last instar nymphs of H. patellifera. Meanwhile, a brief discussion on this structure was conducted in the discussion section. Regarding its specific classification, further research is needed in combination with the imaging results of the projection electron microscope.

Comment 4: Line 124: What is a Bohm bristle- cuticular structure without sensory function, or are they sensilla?. It should be explained because in the description part of the sensilla, Bohm bristles are not described.

Response: Thank for your suggestion. The Bohm bristle cuticular structure belongs to mechanical receptors. Due to our negligence, the description of this sensilla was omitted. The relevant description has been added in the description section of sensilla.

Comment 5: Line 126: Fig. 7c doesn't confirm this description.

Response: Thank you for your careful reading. We made a low-level mistake in the citation of the picture and we are deeply sorry for it. Figure 7c has been deleted in and in the citation of this sentence.

Comment 6: Line 154: What is a type sensilla in contrast to subtypes? According to this division, there are five subtypes (two sensilla trichodea and three campaniformea).

Response:  Thank you for your suggestion. We agree with this comment. In the previous text, the originally named campaniformia sensilla (ScaIII) has been classified as an uncertain cuticular structure. Therefore, all the subtypes in the text have been changed to four.

Comment 7: Line 160: How did the authors recognize the terminal pore, since it is not marked or visible in the photos?

Response 7: We are deeply sorry for this description. This structure is described based on the spike sensor in this literature. After a more detailed ultrastructural observation this time, no porous structure was found at the tip. This description was deleted in the manuscript.

Comment 8: Lines 162-163: Should be subtypes (marked also on the figures). Line 411: Authors should use "subtypes".

Response 8: In this study, the sensilla chaetica (Sc) exhibit continuous variation in length dimensions, and it is difficult to make an accurate distinction among 3 forms. In order to show the diversity of sensilla chaetica (Sc), we defined them as elongate, intermediate, robust form. And other reviewers suggested that it should not differentiated by subtypes.

Comments on the Quality of English Language

Many of the entries in the Results and Discussion sections are very long. There are problems with the use of the singular and plural forms, and it is often unclear whether one or many flagellomeres are being referred to.

Response: We appreciate the reviewer's careful attention to these details. They checked and modified all the singular and plural forms in this part to enhance the correctness of the article.

Reviewer 3 Report (New Reviewer)

Comments and Suggestions for Authors

Insect functional appendages, such as antennae, often exhibit sexual dimorphism due to their role in detecting sex-specific signals (e.g., sex pheromones, environmental volatiles). Investigating antennal sensilla is crucial for understanding morphological adaptations and behavioral traits linked to these sensory structures. This study analyzes the ultrastructure, sensilla types, distribution patterns, and developmental process of antennae in nymphs and adults of Hierodula patellifera using scanning electron microscopy (SEM). The work explores sexual dimorphism, flagellomere addition, and sensilla diversity, advancing our understanding of antennal morphology and development in Mantodea. The experimental design is robust, and the findings are well-supported by SEM imaging. However, some of the methodological details, data presentation, and discussion depth are needed to enhance clarity and impact. Specific recommendations are listed below.

Line 52, the author should briefly link antennal morphology to predatory behavior or habitat adaptation (e.g., how sensilla distribution aids prey detection, or specific behaviors). This may be beneficial for understanding the ecological role of antennal sensilla.

Line 86, please clarify the gender distribution. For instance, 5 individuals per instar should state if the sexes were equally represented.

Line 89, please specify the parameters (e.g., duration, power) for antennal cleaning, as variations could affect delicate nymphal structures and sensillar morphology (e.g., fine structure).

Line 342, error bars are not apparent in figures 7–8.

Line 430-432, The author can expand on the potential roles of hormones (e.g., ecdysone) in regulating antennal growth and sexual dimorphism.

Line 476-483, Please discuss how Sgp density in males correlates with pheromone detection efficiency and mating success in natural habitats. Some new references may be added to this manuscript.

Line 487-505, Please strengthen arguments for using antennal traits in species differentiation by comparing H. patellifera with more Mantidae taxa.

Line 165 Figure 3, Line 184 Figure 4, Line 212, Image Resolution, provide some enhanced resolution in critical SEM images (e.g., Figures, 3f-g; 4f; 5e) to clarify sensilla morphology, if the author has.

Author Response

Reviewer #3 (Comments to the Author):

Insect functional appendages, such as antennae, often exhibit sexual dimorphism due to their role in detecting sex-specific signals (e.g., sex pheromones, environmental volatiles). Investigating antennal sensilla is crucial for understanding morphological adaptations and behavioral traits linked to these sensory structures. This study analyzes the ultrastructure, sensilla types, distribution patterns, and developmental process of antennae in nymphs and adults of Hierodula patellifera using scanning electron microscopy (SEM). The work explores sexual dimorphism, flagellomere addition, and sensilla diversity, advancing our understanding of antennal morphology and development in Mantodea. The experimental design is robust, and the findings are well-supported by SEM imaging. However, some of the methodological details, data presentation, and discussion depth are needed to enhance clarity and impact. Specific recommendations are listed below.

Response : Thank you for your careful reading and invaluable comments, we have conscientiously revised our manuscript following your advice. Specific modifications are as follows.

Comment 1: Line 52, the author should briefly link antennal morphology to predatory behavior or habitat adaptation (e.g., how sensilla distribution aids prey detection, or specific behaviors). This may be beneficial for understanding the ecological role of antennal sensilla.

Response: We agree with this comment. In the introduction part, relevant content was supplemented, briefly introducing that the predation of mantid is not only related to vision, but also relies on its antennae receptors.

Comment 2: Line 86, please clarify the gender distribution. For instance, 5 individuals per instar should state if the sexes were equally represented.

Response: Thank for your suggestion. In the material method, we have added the relevant description that “five male and female individuals of each instar and adult were selected as samples” (line 87).

Comment 3: Line 89, please specify the parameters (e.g., duration, power) for antennal cleaning, as variations could affect delicate nymphal structures and sensillar morphology (e.g., fine structure).

Response: Thank for your suggestion. We have supplemented and improved in the original manuscript that “Nymphs at the 1st to 5th instars should be washed at 45KHz for no more than 2 minutes, while those after the 5th instars need to be washed at the same frequency for 3 minutes” (In lines 93-94).

Comment 4: Line 342, error bars are not apparent in figures 7–8.

Response: We agree with this comment. Changes were made to Figures 7 and 8 to make the error bars in the figures more obvious.

Comment 5: Line 430-432, The author can expand on the potential roles of hormones (e.g., ecdysone) in regulating antennal growth and sexual dimorphism.

Response: We agree with this comment. In the manuscript, we gave examples to illustrate that ecdysone has a certain influence on the sexual dimorphism of insects, and this phenomenon may contribute to the research related to the development of mantids.

Comment 6: Line 476-483, Please discuss how Sgp density in males correlates with pheromone detection efficiency and mating success in natural habitats. Some new references may be added to this manuscript.

Response: We agree with this comment. Some new references and discussion have been added in the relevant sections. The sensilla grooved peg (Sgp) distribution shows obvious sexual dimorphism in adults. Male adults have more of this sensilla than females and appear earlier on the flagellum. These features are common in other mantids [1, 2]. Studies have shown that males with a higher number of Sgp can detect pheromones more quickly, which increases their chances of successful mating.

[1] Carle, T.; Toh, Y.; Yamawaki, Y.; Watanabe, H.; Yokohari, F. The antennal sensilla of the praying mantis Tenodera aridifolia: A new flagellar partition based on the antennal macro-, micro- and ultrastructures. Arthropod Struct. Dev. 2014, 43, 103-116.

[2] Jayaweera, A.; Barry, K.L. Male antenna morphology and its effect on scramble competition in false garden mantids. Natur- wissenschaften 2017, 104, 75.

Comment 7: Line 487-505, Please strengthen arguments for using antennal traits in species differentiation by comparing H. patellifera with more Mantidae taxa.

Response: The morphological characteristics of antennae can be used to distinguish advanced taxonomic mediators, and there is little difference among similar species. There are significant differences among different families, and the differences occur in the external morphology of the antennae. All flagellomere in males of H. patellifera (Mantidae) is cylindric with equal width, while the flagellomere of C. nebulosa (Hymenopodidae) is cone shaped nearly, contracted at the proximally and become wider distally. The sexual dimorphism in the length of antennae C. nebulosa is more remarkable than in H. patellifera. Furthermore, The grooved peg sensilla of C. nebulosa appeared earlier on the flagellum, which makes the two different in “The segmentation of the flagellum”. Therefore, the antennae external morphology, types and distribution of sensilla of antennal sensilla could be diagnostic features among family of Mantodea. The relevant content has also been stated in the discussion section (In lines 496-526).

Comment 8: Line 165 Figure 3, Line 184 Figure 4, Line 212, Image Resolution, provide some enhanced resolution in critical SEM images (e.g., Figures, 3f-g; 4f; 5e) to clarify sensilla morphology, if the author has.

Response: We agree with this comment. We have replaced the pictures you mentioned in the text as much as possible.

Reviewer 4 Report (New Reviewer)

Comments and Suggestions for Authors

The paper is interesting and the research is valuable to help understant the biology of mantids. The authors did a good job presenting the variations that are present from 1st instar nymph to adults.

Some points that must be improved are the quality of the images. Part of the title of the paper is "Ultrastructure of the Antennal Sensilla". However, most sensilla are presented with low magnification, which does not allow the reader to see the fine details present, such as the presence or absence of pores, ornamentation, etc. For instance, sensilla basiconica are typically multiporous, but in the paper, no pores can be distinguished due to the low magnification. Sensilla 4c-d  and 5c-e are the only ones with adequate resolution. The rest must be improved. I kindly suggest the authors review the following paper:

https://www.researchgate.net/profile/At_Rani/publication/356622133_Morphological_characterization_and_distribution_of_antennal_sensilla_of_Helicoverpa_armigera_Hubner_Lepidoptera_Noctuidae_using_scanning_electron_microscopy/links/634559852752e45ef6ad4576/Morphological-characterization-and-distribution-of-antennal-sensilla-of-Helicoverpa-armigera-Huebner-Lepidoptera-Noctuidae-using-scanning-electron-microscopy.pdf

Supplementary tables S1 and S2 seem to be missing, or at least I couldn't find them. The information presented in the tables seems valuable enough to be included in the main text.

The abstract and the results should be written in the past tense.

The title sound a little awkward, especially the last part (with Discussion of its Flagellomeres Addition). I suggest the authors consider changing the title to something similar :

Antennal Sensilla Ultrastructure and Flagellomere Addition in Nymphs and Adults of Hierodula patellifera (+ author, family etc)

Ultrastructure of Antennal Sensilla in Nymphs and Adults of Hierodula patellifera, with a Discussion of Flagellomere Addition (+ author, family etc)

I have provided additional comments in the attached PDF file.

In my opinion, the research presented is valuable and worth publishing after some major chages.

Comments on the Quality of English Language

There are numerous errors in the text. Although they do not particularly hinder comprehension, they should be corrected to improve clarity, readability, and proffessionalism.

Author Response

Reviewer 4:

The paper is interesting and the research is valuable to help understant the biology of mantids. The authors did a good job presenting the variations that are present from 1st instar nymph to adults.

Response:

Thank you for your careful reading and invaluable comments, we have conscientiously revised our manuscript following your advice. Specific modifications are as follows.

Comment 1: Some points that must be improved are the quality of the images. Part of the title of the paper is "Ultrastructure of the Antennal Sensilla". However, most sensilla are presented with low magnification, which does not allow the reader to see the fine details present, such as the presence or absence of pores, ornamentation, etc. For instance, sensilla basiconica are typically multiporous, but in the paper, no pores can be distinguished due to the low magnification. Sensilla 4c-d, and 5c-e are the only ones with adequate resolution. The rest must be improved. I kindly suggest the authors review the following paper:

Response:

We agree with this comment. For the low magnification pictures in the article, we re-shoot them to enable readers to see a more detailed structural presentation.

Comment 2: Supplementary tables S1 and S2 seem to be missing, or at least I couldn't find them. The information presented in the tables seems valuable enough to be included in the main text.

Response:

Appendix S1 and S2 are generally quite long and take up a relatively small proportion of the content in the article. Therefore, they are placed as appendices in the article to enhance its aesthetic appeal. Meanwhile, the table has been attached at the end of the manuscript.

Comment 3: The abstract and the results should be written in the past tense.

Response:

Thank you for your suggestion. The tenses of both the abstract and the result sections in the article have been changed to the past tense.

Comment 4: The title sound a little awkward, especially the last part (with Discussion of its Flagellomeres Addition). I suggest the authors consider changing the title to something similar :

Antennal Sensilla Ultrastructure and Flagellomere Addition in Nymphs and Adults of Hierodula patellifera (+ author, family etc)

Ultrastructure of Antennal Sensilla in Nymphs and Adults of Hierodula patellifera, with a Discussion of Flagellomere Addition (+ author, family etc)

Response:

We agree with this comment. The title of the article has been changed “Antennal Sensilla Morphology and Flagellomere Addition in Nymphs and Adults of Hierodula patellifera Serville, 1839 (Mantodea: Mantidae)”.

Comments on the Quality of English Language

There are numerous errors in the text. Although they do not particularly hinder comprehension, they should be corrected to improve clarity, readability, and professionalism.

Response:

We sincerely thank the reviewers for their careful reading and valuable feedback on the language errors in the manuscripts. The errors in the text have been corrected to ensure the professionalism and readability of the article.

Comments in the attached PDF file.

Comment 1: Here it seems to say that you carried out SEM studies of other insects, which is not the case. Perhaps you should make it clear that you only discuss the similarities and differences.

Response:

We agree this comment. The content in the text has been changed to "the similarities and differences were compared with other insects".

Comment 2: Please clarify this. It seems to say that the number of antennae vary according to sex?

Response 2:

This sentence in the manuscript has been changed to “Antennae may vary considerably in length, morphology, type, number of segmentation and the size in different sexes of the same species”.

Comment 3: If I understood correctly, these parts were classified according to sensillar patterns that were present in those flagellomeres but not in others. If so, please emphasize and explicitly say so. Please emphasize that these are the characteristics that you used to classify the different parts. I think this explanation (from line 267 to 289) must be written before the previous section that explains the different parts (243-266). This would make everything much clearer.

Response:

In order to represent these variations precisely, we divided the flagellum of adults into six regions based on the external morphology and distributional features of sensilla, from the most proximal (part I) to the most distal (part VI). The contents described in these different parts (part I to part VI) are based on the appearance of different types of sensilla as relevant evidence.

Comment 4: what does "meristal" mean?

Response:

In the manuscript, "meristal segments" indicates that the meriston divides into multiple flagellomeres.

Others comments:

Response:

The issues of word expression and pictures mentioned in the pdf comments have all been modified.

Round 2

Reviewer 4 Report (New Reviewer)

Comments and Suggestions for Authors

The paper improved markedly after the changes. It still has minor errors in the language that I have marked in the PDF file.

As a side note, the use of vector graphics instead of bitmap images would enhance the overall quality of the document, particularly in figures containing charts and graphs. Vector images can be scaled without loss of resolution, allowing for clearer visualization when zoomed in. In particular, Figures 7, 8, and 10 would benefit from being provided as vector graphics. This is merely a suggestion and not a requirement. The final decision rests entirely with the authors.

Comments on the Quality of English Language

Just minor errors that I have maked in the attached PDF file

Author Response

Comment 1: The paper improved markedly after the changes. It still has minor errors in the language that I have marked in the PDF file.

Response:

Thank you for your careful review of the manuscript and for identifying the

minor errors in the attached PDF file. These errors have been corrected in the revised manuscript

Comment 2: As a side note, the use of vector graphics instead of bitmap images would enhance the overall quality of the document, particularly in figures containing charts and graphs. Vector images can be scaled without loss of resolution, allowing for clearer visualization when zoomed in. In particular, Figures 7, 8, and 10 woul benefit from being provided as vector graphics. This is merely a suggestion and not a requirement. The final decision rests entirely with the authors.

Response:

We thank the reviewer for this helpful suggestion. We agree that vector graphics will improve the clarity and scalability of figures in the manuscript. However, our individual graphics are displayed in the software in the form of vector graphics. After our splicing, they ultimately cannot be presented in the form of vector graphics.

Therefore, we still submit them in the form of regular graphics. On this basis, we have improved the resolutions of Figures 7, 8 and 10 to enhance the overall clarity.

Comments on the Quality of English Language

Comment: Just minor errors that I have made in the attached PDF file

Response:

Thank you for your careful review of the manuscript and for identifying the

minor errors in the attached PDF file. These errors have been corrected in the revised manuscript.

This manuscript is a resubmission of an earlier submission. The following is a list of the peer review reports and author responses from that submission.

Round 1

Reviewer 1 Report

Comments and Suggestions for Authors

1. I suggest that the authors check the statistics they employed for their analysis

2. Add a discussion in terms of what they found in their statistical results (check point 1 please)

3. Consider using the traditional terminology of antennae parts or modify it for a better scientific terminology

4. Review and correct all observation marked along the text including the literature cited please. 

Author Response

We appreciate your time in reviewing our manuscript. Detailed responses are provided below, along with the revised and corrected files.

Comment 1: I suggest that the authors check the statistics they employed for their analysis

Response:

We checked the statistics seriously, and compared with several publications. The method you proposed is indeed a good one, it is relatively suitable for comparison between each two groups or small number of groups. There are 14 groups data in our study, which is relatively large. In addition, our data needs to be compared between two or more groups simultaneously. If calculated our data set by the Kruskal-Wallis method, the significance difference will be complicated. In the statistical process, turkey method is used for post hoc comparison, which not only reduces the occurrence of Type I Error, but also determines whether there is a significant difference between two or more groups. Therefore, we think the one-way analysis of variance is suitable for our work. This method is also employed in some related researches as follows:

(1) Yang, H.-Y.; Zheng, L.-X.; Zhang, Z.-F.; Zhang, Y.; Wu, W.-J. The structure and morphologic changes of antennae of Cyrtorhinus lividipennis (Hemiptera: Miridae: Orthotylinae) in different instars. PLOS ONE 2018, 13, e0207551.

(2) Zhu, W.; Yang, L.; Long, J.; Chang, Z.; Mu, Y.; Zhou, Z.; Chen, X. Morphology of the antennal sensilla of the nymphal instars and adults in Notobitus meleagris (Hemiptera: Heteroptera: Coreidae). Insects 2023, 14, 351.

Comment 2: Add a discussion in terms of what they found in their statistical results (check point 1 please).

Response:

We agree with this comment. According to the overall structure of the paper and the statistical results. We removed the first point of discussion in original manuscript, and added the new Discussion.

Comment 3: Consider using the traditional terminology of antennae parts or modify it for a better scientific terminology.

Response:

Traditional terminology proximal, medial, and distal, which refer to imprecise longitudinal regions. The antennae of praying mantis are filiform, long and complicated, with more than 100 flagellomere. In order to represent the variations and diversity precisely. We adopted a new terminology aims to make the mantid antennae more specific. This new term is proposed by referring to Carle et al entitled “The antennal sensilla of the praying mantis Tenodera aridifolia: a new flagellar partition based on the antennal macro-, micro- and ultrastructures.”

In addition, apologize for causing confusion to the review. In the original manuscript: “we divide the flagellum of adults into six regions, from the most proximal (part I) to the most distal (part VI).” This sentence is just a direction-pointing question, the part I to part VI is start from the proximal. It does not mean that “Part I” is replaced with the “most proximal” and “Part VI” is replaced with “the most distal”.

Comment 4: Review and correct all observation marked along the text including the literature cited please. 

Response:

Thank you for your detailed revisions of our manuscript. We agree with this comment. The details regarding language expression and other issues have been revised one by one. The content has been modified based on the feedback received.

Comments in the attached file.

Comment 1: Please, specify location and identification of what?

Response:

We agree with this comment. We have revised this sentence as following: “Insect antennae play an important role in the process of information exchange such as host location, host discrimination, habitat searching, avoiding danger, mating and reproduction, and pheromone acceptance”.

Comment 2: All insects have a pair of antennae

Response:

We agree with this comment. All insects have a pair of antennae, the “Generally” is improper statement. In order to make the Introduction section of the article more coherent, we removed this sentence and rewrote the Introduction section.

Comment 3: What´s the relation between the cuticle hardness and the need to clean the specimens? clarify this point please.

Response:

We agree with this comment. We have revised this sentence as following: “Washing 1-3 minutes with an ultrasonic cleaner (KQ-250DM, KunShan Ultrasonic Instruments Co., Ltd, Jiangsu, China). The cuticular structures of low age nymphs is more delicate than the adults and old mature nymphs, so the duration of washing varied depending on the instar.”

Comment 4: For experiments with few data it is not recommended to use parametric statistics, instead, it would be much better to use non-parametric ones. In this case a Kruskal Wallis would be more appropriate. The authors should consider this very important factor, due to it might yield new and relevant information.

Response:

We checked the statistics seriously, and compared with several publications. The method you proposed is indeed a good one, it is relatively suitable for comparison between each two groups or small number of groups. There are 14 groups data in our study, which is relatively large. In addition, our data needs to be compared between two or more groups simultaneously. If calculated our data set by the Kruskal-Wallis method, the significance difference will be complicated. In the statistical process, turkey method is used for post hoc comparison, which not only reduces the occurrence of Type I Error, but also determines whether there is a significant difference between two or more groups. Therefore, we think the one-way analysis of variance is suitable for our work. This method is also employed in some related researches as follows:

(1) Yang, H.-Y.; Zheng, L.-X.; Zhang, Z.-F.; Zhang, Y.; Wu, W.-J. The structure and morphologic changes of antennae of Cyrtorhinus lividipennis (Hemiptera: Miridae: Orthotylinae) in different instars. PLOS ONE 2018, 13, e0207551.

(2) Zhu, W.; Yang, L.; Long, J.; Chang, Z.; Mu, Y.; Zhou, Z.; Chen, X. Morphology of the antennal sensilla of the nymphal instars and adults in Notobitus meleagris (Hemiptera: Heteroptera: Coreidae). Insects 2023, 14, 351.

Comment 5: Figure 3d and 3e requires to be pointed out

Response:

We agree with this comment. In page 6, Figure 3d and 3e were changed and corrected.

Comment 6: require to be pointed out in the figure

Response:

In page 7, Figure 4a was changed and corrected.

Comment 7: It is not pointed out in the figure

Response:

We agree with this comment. In page 8, Figure 5d and 5e was changed and corrected.

Comment 8: I understand that, the six divisions are Part I to VI, but it is very confusing now with the divisions a-l. Please clarify all this the 3rd instar in figure 6 is not explained in the text, besides, it corresponds to male or female?

Response:

We agree with this comment. We have added explanations for figures a-l in our manuscript as following (Page 10, Lines 309-312): “SEM images (Figure a, c, e, g, i, k) represent six parts of adult males, while SEM images (Figures b, d, f, h, j, l) represent six parts of adult females. SEM images (Figure a1, c1, e1, g1, i1) represent six parts of 9th-instar males, while SEM images (Figures b2, d2, f2, h2, j2) represent six parts of 9th-instar females.” In page 9, line 285-286, we have added “3rd -instar nymph”.

Sexual dimorphism could not be observed from the external morphology of genitalia in 3rd instar, and the distribution of sensilla patterns does not varied along the longitudinal axis of the antennae and between sexes. Therefore, the gender cannot be distinguished at 3rd instars. We have also wrote this in the Materials and Methods section in our manuscript as following: “From the 4th instar, the gender of nymphs can be distinguished by their genitalia.” (In page 2, line 80).

Comment 9: Authors suggested a new terminology, but it seems they are not applying it. Please clarify if the new terminology will be applied or they will use the traditional terminology. I suggest to use the traditional terminology.

Response:

Traditional terminology proximal, medial, and distal, which refer to imprecise longitudinal regions. The antennae of praying mantis are filiform, long and complicated, with more than 100 flagellomere. In order to represent the variations and diversity precisely. We adopted a new terminology aims to make the mantid antennae more specific. This new term is proposed by referring to Carle et al entitled “The antennal sensilla of the praying mantis Tenodera aridifolia: a new flagellar partition based on the antennal macro-, micro- and ultrastructures.”

In addition, apologize for causing confusion to the review. In the original manuscript: “we divide the flagellum of adults into six regions, from the most proximal (part I) to the most distal (part VI).” This sentence is just a direction-pointing question, the part I to part VI is start from the proximal. It does not mean that “Part I” is replaced with the “most proximal” and “Part VI” is replaced with “the most distal”.

Comment 10: check redaction please

Response:

We agree with this comment. In page 11, Line 353. We change “increases successive instars” to “increases in successive instars”.

Comment 11: The standard deviation is not explained

Response:

We apologize for causing confusion to the review. The standard deviations from our experimental results have been indicated in each figure. Figures A-D show them within the circles, while Figures E-F display them above the bar charts.

Comment 12: check redaction please

Response:

We agree with this comment. In page 16, Lines 406-408. We changed “This diagram shows a part of flagellomeres newly formed during molting is generated at the proximal end of antennae by the division of the meriston” to “This diagram shows a part of flagellomeres, newly formed during molting, is generated at the proximal end of antennae by the division of the meriston.”

We have studied all comments meticulously, and improved the manuscript. All revisions to the manuscript are highlight in red.

Reviewer 2 Report

Comments and Suggestions for Authors

This paper thoroughly decribes the antennal morphology of the Nymphs and Adults of mantids Hierodula patellifera. I find that it’s significantly improved since the last version, however, I am afraid this does not meet the quality of publication in Insects, the Introduction and Discussion still need rewriting, and the Results requires a substantial amount of editing. Some major issues:

Poorly written Introduction, did not justify this study, or what is the scientific question this study aims to answer.

Wrong statistics practice, corrections are needed for multiple comparative analysis (see my comments).

Presentations of Figures 7, 8, and 9 need improving, the results from these figures need to be presented clearly.

Wordy Discussion that largely repeated the Results, making the key points rather difficult to find, some points are not fully supported by the references cited.

Please refer to the comments in the comments in the attached file.

Comments on the Quality of English Language

Significantly improved, although try to improve the conciseness and cleaness.

Author Response

This paper thoroughly describes the antennal morphology of the Nymphs and Adults of mantids Hierodula patellifera. I find that it’s significantly improved since the last version, however, I am afraid this does not meet the quality of publication in Insects, the Introduction and Discussion still need rewriting, and the Results requires a substantial amount of editing. Some major issues:

Response:

Thank you for your detailed revisions of our manuscript. We agree with this comment. Regarding the language expression and other issues, we studied the comments meticulously and improved the manuscript carefully. The content has been modified based on the feedback received. All revisions to the manuscript are highlight in red. Specific modifications are as follows.

Comment 1: Poorly written Introduction, did not justify this study, or what is the scientific question this study aims to answer.

Response:

We agree with this comment, according to your advice, we rewrote the Introduction section. Specific modifications are as follows: “We first demonstrate the importance of antennae in revealing the sexual dimorphism in insects. The second part presents research on sexual dimorphism in mantids and highlights the shortcomings in the study of mantid antennae. In the last part, we briefly introduce the significance of the selection of H. patellifera for this study and its contribution to future research.”

Comment 2: Wrong statistics practice, corrections are needed for multiple comparative analysis (see my comments).

Response:

We checked the statistics seriously, and compared with several publications. There are 14 groups data in our study, which is relatively large. In addition, our data needs to be compared between two or more groups simultaneously. In the statistical process, turkey method is used for post hoc comparison, which not only reduces the occurrence of Type I Error, but also determines whether there is a significant difference between two or more groups. In addition, the purpose of statistical data is to verify that the most obvious variation of mantid in the whole antennal development process occurred from the 5th to 6th instar. The results show that the most significant variation occurs in the 6th instar, and the antennae length of 6th-instar nymphs is about 1.25 times as long as 5th-instar nymph (The length of the scape and pedicel also reflect this). The results support this conclusion. Therefore, we think the one-way analysis of variance is suitable for our work. This method is also employed in some related researches as follows:

(1) Yang, H.-Y.; Zheng, L.-X.; Zhang, Z.-F.; Zhang, Y.; Wu, W.-J. The structure and morphologic changes of antennae of Cyrtorhinus lividipennis (Hemiptera: Miridae: Orthotylinae) in different instars. PLOS ONE 2018, 13, e0207551.

(2) Zhu, W.; Yang, L.; Long, J.; Chang, Z.; Mu, Y.; Zhou, Z.; Chen, X. Morphology of the antennal sensilla of the nymphal instars and adults in Notobitus meleagris (Hemiptera: Heteroptera: Coreidae). Insects 2023, 14, 351.

Comment 3: Presentations of Figures 7, 8, and 9 need improving, the results from these figures need to be presented clearly.

Response:

We agree with this comment, according to your advice, we have revised presentations of Figures 7, 8, and 9. First, in order to present the experimental results more clearly, E and F in Figure 7 and Figure 8 in the original manuscript are corrected. We represent instar and gender on the X-axis in figures 7 and 8 as following: “The number represents instar in Figure C and D, a and b indicate gender. a indicates males, b indicates females”. In addition, Change the overall description of Figure 9 to “SEM images represent the sections of two growth sites derived from H. patellifera.”. This will help us better understand the content of Figure 9.

Comment 4: Wordy Discussion that largely repeated the Results, making the key points rather difficult to find, some points are not fully supported by the references cited.

Response:

We agree with this comment. In order to make the discussion points in the article more prominent. We removed the first point of discussion in original manuscript, and added the new Discussion.

Comments in the attached file.

Comment 1: Did not mention how you construct the developing schematics of antennae (Fig. 10).

Response:

We agree with this comment, according to your advice, we have added this construct the developing schematics of antennae (Fig. 10) in the Materials and Methods section as following: “we develop a schematics of addition of flagellomere. This model uses the number 5 to represent the number of flagellomeres in each part from Parts II to VI, illustrating growth patterns of two growth sites.” (see page 3, lines 112-115).

Comment 2: Which is it? 20Kv is quite high for delicate structures. Please double check.

Response:

We agree with this comment. We have corrected "7-20kV" into "7-10kV".

Comment 3: Not sure what information we can get from this figure other than the antennae of patellifera are filiform, and they increase as they grow, which we already know. This figure seems a bit redundant.

Response:

We agree with this comment. We have removed 5th and 9th instars from Figure 2 to reduce the length of the figure, but retained an important part of our experimental results. Figure 2 indicates that the key point for the emergence of sexual dimorphism occurs during the 7th instar. In addition, this Figure can better reflect the sexual dimorphism of antennae visually, so we want to keep this Figure 2.

Comment 4: Wrong statistic practice. For multiple comparisons like this, you should do corrections to avoid false positive results. These levels are meaningless in describing insect antennae: we all know insect antennae increase as they grow, even though thy did not show statistic difference between instars. Think about what question the statistic test could answer.

Response:

We checked the statistics seriously, and compared with several publications. There are 14 groups data in our study, which is relatively large. In addition, our data needs to be compared between two or more groups simultaneously. In the statistical process, turkey method is used for post hoc comparison, which not only reduces the occurrence of Type I Error, but also determines whether there is a significant difference between two or more groups. In addition, the purpose of statistical data is to verify that the most obvious variation of mantid in the whole antennal development process occurred from the 5th to 6th instar. The results show that the most significant variation occurs in the 6th instar, and the antennae length of 6th-instar nymphs is about 1.25 times as long as 5th-instar nymph (The length of the scape and pedicel also reflect this). The results support this conclusion. Therefore, we think the one-way analysis of variance is suitable for our work. This method is also employed in some related researches as follows:

(1) Yang, H.-Y.; Zheng, L.-X.; Zhang, Z.-F.; Zhang, Y.; Wu, W.-J. The structure and morphologic changes of antennae of Cyrtorhinus lividipennis (Hemiptera: Miridae: Orthotylinae) in different instars. PLOS ONE 2018, 13, e0207551.

(2) Zhu, W.; Yang, L.; Long, J.; Chang, Z.; Mu, Y.; Zhou, Z.; Chen, X. Morphology of the antennal sensilla of the nymphal instars and adults in Notobitus meleagris (Hemiptera: Heteroptera: Coreidae). Insects 2023, 14, 351.

Comments 5: not sure are those box plots inside the circles? If they are, indicate the what each quantile means. Very hard to understand. I don't think the exact width or length of the each antennal flagellomere is that important taxonomy characters, admire the effort, but don't think we need to report. 

Response:

The content in the circle does not represent the box plots and it's the standard deviation of the data. By measuring the width and length of the flagellomeres, we found that a significant sexual dimorphism begins to appear during the 7th instar. Figure 7 indicate that: Until the 7th instar, the width of the flagellomeres has an obvious distinction between the sexes (Figure 7A). In contrast, after the 6th instar and adults, males evidently bear wider flagellomeres than females, except the terminal part of the antennae (Figures 7A-B). In addition, by measuring the width of the first flagellomere, until the 7th instar, the width of the first flagellomere has an obvious distinction between the sexes. We investigated the differences in width of flagellomeres at the distal parts of antennae from 1st instar to adult by averaging the width between the 10th and 30th flagellomeres via beginning to calculate from the antennae tip. The results showed that the width of flagellomeres has obvious differences between the sexes at the 7th instar. These data further supports the reliability of this result.

Comment 6: Not clear why the flagellomeres need to be measured every 10 segments and every segments. It's essentially the same information, however, not understand why the patterns in for example A and B, C and D are different? Am I missing anything?

Response:

We agree with this comment. Due to the longer antennae length of the adult insects, we chose to measure the length every ten segments to reflect the changes in the length of antennae (Figure 8A). For the nymphs, we selected to measure the length every five segments to clearly reflect the changes in the length of antennae. The length of flagellomeres of other instars are similar, only data for antennae of 9th instar antennae are shown here (Figure 8C). The Figure 8A and 8C indicate that: We found that the length increased near the first flagellomere (Part I) through length measurements, proving the presence of a growth point in that area. Additionally, The curves also reflects the existence of a growth point between Part II and Part III, confirming that the increase in length of Part II compensates for the decrease in length of Part III, which is consistent with the results shown in Figure 10. At the same time, to verify the authenticity of the first growth site, we measured the antennae lengths of both adults and nymphs at every segment (Figure 8B, C). It is found that the length of the flagellum gradually increases in the Part I and Part II, which further proves that the antennae has a point at the position of the first flagellomere and the flagellomere is added.

Comment 7: In addition, captions way too small to read and the figures are very difficult to understand. Please use other method, patterns or captions on X axis to indicate the instars or sexes.

Response:

We agree with this comment. In order to present the experimental results more clearly, E and F in Figure 7 and Figure 8 in the original manuscript are corrected. We represent instar and gender on the X-axis in figures 7 and 8 as following: “The number represents instar in Figure C and D, a and b indicate gender. a indicates males, b indicates females” (see page 12 and 13, lines 361-363 and 373-374).

Comment 8: This is not a diagram, although it is very helpful if you could provide a diagram: difficult to understand.

Response:

We agree with this comment. We have rewrote the sentence in the revised manuscript as following “SEM images represent the sections of two growth sites derived from H. patellifera”.

Comment 9: Rather lengthy and quite difficult to comprehend, thus obscuring the most important points of your study. You don't need to discuss everything you found. In a paper like this, you just need to discuss 3-4 points that you think are most important.

Response:

We agree with this comment. In order to make the discussion points in the article more prominent. We removed the first point of discussion in original manuscript, and rewrote the Discussion focus on the sexual dimorphism and comparison of antennae features with other insects.

Comments 10: Since this relationship has already been investigated in "numerous studies". why is this important to illustrate again? In addition, probably this most interesting point of this discussion, but poorly arranged. Lines 515-522 basically repeated your results. You can focus more on the implications of the growth pattern you discovered, but what mentioned in lines 522-524 is not one of them.

Response:

We agree with this comment, according to your friendly suggestion, we rewrote this paragraph, and added discussion content about the second partial division as following: “The antennal development of termites and cockroaches has been studied in recent years. In the antennae of termites and cockroaches, the primary flagellomeres divide once, producing two secondary flagellomeres each [25,26]. In contrast, the number of new flagellomeres formed from the first flagellomere is irregular in H. patellifera antennae, but has the same way to add the flagellomeres at this point. However, the first flagellomere consistently generates a single new meristal segment at each molt in grasshoppers and mantophasmids [27,28]. In addition, the H. patellifera antennae have a second site to add the flagellomeres. A single flagellomere from the secondary partial site regularly divides into two flagellomeres, thereby increasing the number of distal partial flagellomeres. Although a secondary site for antennae growth was also previously discovered in grasshoppers during their early development, this pattern is different from the mantids [28,29]. In grasshoppers, the secondary site divides into multiple flagellomeres at a given molt [30]. Antennal development in insects illustrates that mantids have a closer relationship with termites and cockroaches, whereas they are more distantly related to Orthopteran species.” (see page 17, lines 484-497).

Comments 11: Language, too general, not specific enough.

Response:

We agree with the comment. We have rewrote this sentence in the revised manuscript as following “However, the sensilla type in adults between mantids and cockroaches has some differences.”

Comments on the Quality of English Language

Comment: Significantly improved, although try to improve the conciseness and cleanliness.

Response:

We agree with this comment. We have now worked on both language and readability and have also involved native English speakers for language corrections.

Reviewer 3 Report

Comments and Suggestions for Authors

Dear Authors,

I appreciate the work you have put into improving the manuscript. However, it still has some shortcomings. I have noted minor comments directly in the text.

In my opinion, the most serious problem is the issues raised in the chapter "Types and Morphology of Sensilla in the Nymphs and Adults". It is still unknown where the authors got the name of the sensilla from, and there are no large-magnification illustrations of individual sensilla.

Without detailed analyses of the sensilla structure (porous/non-porous, surface structure), sensilla cannot be adequately classified (unless we have TEM data). Size and shape alone do not allow for drawing proper conclusions. Sensilla, especially from the "long" group, mainly mechanoreceptors, are often differentiated by length within one type. For example, I do not see a difference between sbI and st II. Both the nests and the surface are poorly imaged.

In summary, although the manuscript has been significantly improved, it still requires work on nomenclature and correct classification of sensory organs.

Comments on the Quality of English Language

The language is fully understandable, but its fluency could be improved.

Author Response

I appreciate the work you have put into improving the manuscript. However, it still has some shortcomings. I have noted minor comments directly in the text.

Response:

Thank you for your careful reading and invaluable comments, we have conscientiously revised our manuscript following your advice. Specific modifications are as follows.

Comment 1: In my opinion, the most serious problem is the issues raised in the chapter "Types and Morphology of Sensilla in the Nymphs and Adults". It is still unknown where the authors got the name of the sensilla from, and there are no large-magnification illustrations of individual sensilla.

Response:

We agree with this comment, according to your friendly suggestion, we have added the the terminology for the antennal sensilla in the revised manuscript (Page 3 and Lines 109-110) as following: “The terminology for the antennal sensilla follows that of Carle et al, and Drilling [2,16].”

Comment 2: Without detailed analyses of the sensilla structure (porous/non-porous, surface structure), sensilla cannot be adequately classified (unless we have TEM data). Size and shape alone do not allow for drawing proper conclusions. Sensilla, especially from the "long" group, mainly mechanoreceptors, are often differentiated by length within one type. For example, I do not see a difference between sbI and st II. Both the nests and the surface are poorly imaged.

Response:

We agree with this comment. In the course of our experiment, we did not observe subtypes of Sb. The only type that exists is distinguished from the St as follows: The tip of the tapered sensillum is slightly blunter than that of two subtypes of St. In addition, the base of Sb is wider than that of two subtypes of St. A tapered sensilla subtype added in subsequent experiments was slightly wider at the end than subtypes of St, but overall similar to StII. After consideration, we ultimately decided to classify SbI as StII, while not categorizing Sb. In future studies, we will focus on this sensilla.

Comment 3: In summary, although the manuscript has been significantly improved, it still requires work on nomenclature and correct classification of sensory organs.

Response:

We agree with this comment, in response to the two points you raised, we have tried our best to make revisions in our manuscript.

Reviewer #3: Comments in the attached file.

Comment 1: In all cases there should be a comma between the name and the year.

Response:

We agree with this comment, according to your advice, we have added a comma between the name and the year. Such as: “Hierodula patellifera Serville, 1839 (Mantodea: Mantidae)”.

Comment 2: Keywords should not repeat words used in the title.

Response:

We agree with this comment, according to your advice, we have replaced the keyword “Hierodula patellifera” with “antennal development”.

Comment 3: Please In addition, this information should be included in the section called "terminology".

Response:

We agree with this comment, according to your advice, we have indicated where the nomenclature of individual types of sensilla was taken from. In addition, we have included everything related to the sensilla in the section called "Terminology" as following: “Identification of various types of sensilla based on the external morphology, length, and distribution [14,15]. The terminology for the antennal sensilla follows that of Carle et al, and Drilling [2,16]. The abbreviation for the sensilla are as follows: sensilla trichodea: St; sensilla basiconica: Sb; sensilla chaetica: Sc; sensilla coelocapitula: Sco; sensilla campaniformea: Sca; sensilla grooved peg: Sgp; Böhm bristle: Bb. In addition, we develop a schematics of addition of flagellomere. This model uses the number 5 to represent the number of flagellomeres in each part from Parts II to VI, illustrating growth patterns of two growth sites. ” (see 2.4 in the Materials and Methods section).

Comment 4: Evolution does not occur for any purpose. It is obvious that this particular set of sensilla is beneficial. Please delete this sentence. It does not contribute anything.

Response:

We agree with this comment, according to your advice, we have deleted this sentence.

Comments on the Quality of English Language

Comment: The language is fully understandable, but its fluency could be improved

Response:

We agree with this comment. We thank the reviewer for this valuable suggestion, and the entire manuscript has been edited by a native English speaker.

Round 2

Reviewer 1 Report

Comments and Suggestions for Authors

The authors have, undoubtedly, improved the paper. I congratulate them for the effort.

However, I found still some pitfalls in the text that can be improved. For example, the annotations related to images are still confusing.

There is some confusion with respect to statistics. It was advised to ensure the normality of the data. If data are distributed normally, then the anova and tukey analysis proceed, otherwise it is recommended to use non-parametric ones. 

Comments on the Quality of English Language

There are a few recommendations in the odf

Author Response

The authors have, undoubtedly, improved the paper. I congratulate them for the effort.

We again appreciate your time in reviewing our manuscript. Detailed responses are provided below, along with the revised and corrected files.

Comment 1: However, I found still some pitfalls in the text that can be improved. For example, the annotations related to images are still confusing.

Response:

We agree with this comment. We have gone through all the images-related annotations in the article and corrected the problematic annotations.

Comment 2: There is some confusion with respect to statistics. It was advised to ensure the normality of the data. If data are distributed normally, then the anova and tukey analysis proceed, otherwise it is recommended to use non-parametric ones. 

Response:

We agree with this comment. The data has been previously tested with reference to the previous literature (1). The specific steps are as follows: Kolmogorov-Smirnov tests were used to test the normality of data and Levene tests were used to determine the homogeneity of variance. Data conforming to the normal distribution and showing homogeneity of variance were analyzed using one-way analysis of variance, and Tukey’s method was used for multiple comparisons. Detailed steps have been added to the Materials and Methods section in our manuscript.

  • Zhu, W.; Yang, L.; Long, J.; Chang, Z.; Mu, Y.; Zhou, Z.; Chen, X. Morphology of the antennal sensilla of the nymphal instars and adults in Notobitus meleagris (Hemiptera: Heteroptera: Coreidae). Insects 2023, 14,

Comments in the attached file.

Comment 1: delete

Response:

We agree with this comment. We have deleted these words “and, are and in both” (see page 1, line 22, 26).

Comment 2: males

Response:

We agree with this comment. We have changed “male” to “males”.

Comment 3: This phenomenon contributes to helping males

Response:

We agree with this comment. We have revised this sentence as following: “This phenomenon helps males to detect sex pheromones” .

Comment 4: The annotations on images are still confusing. I can´t distinguish were are Sb in (g).

Response:

We agree with this comment. We are deeply sorry for our carelessness. We have corrected the annotations and checked all annotations in the article.

Comment 5: The Bb appear in 4g.

Response:

We agree with this comment. We are deeply sorry for our carelessness and have corrected Bb's annotation.

Comment 6: It is not imprecise, but relative, i. e., a distal character could be proximal with respect to another

Response:

In order to more accurately describe the characteristics of antennal flagellomere part, the flagellum were divided into six parts in this study.

Comment 7: Then, under the criteria, 12, 13 or 15, 16 are valid segments to take into account, or it is strictly 1-14?

Response:

The flagellomere #14 is not consistently fixed. During experimental observations, the sensilla grooved peg may or may not present at the distal end of flagellomere #14, so it can be used as both the end of the Part I and the starting point of the Part II. Additionally, in subsequent sections, no repeated flagellomeres are fixed. For example, in the Part II, female still has only sensilla chaetic in flagellomeres #17 to 18, but in the Part III, both sexes suddenly decrease in length from flagellomere #19. The division of these two parts has obvious characteristics.

Comments on the Quality of English Language

There are a few recommendations in the pdf.

Response:

We agree with this comment. We have corrected the language problems mentioned in the pdf.

Reviewer 2 Report

Comments and Suggestions for Authors

This version is much better than the original one, reflecting the input of the authors in improving this paper. I am happy for the authors to go ahead without my involvement. Apologies on my comments about the Tukey’s test, however, I would suggest either improve the table aesthetically, or turn it into a figure to better make your points. Table 1 at this state is a bit hard to read. Specific points:

Line 425. This is not substantiated. This paper only classified the sensilla by morphology, but numerous previous studies shown that functionally different sensilla not necessarily show morphological differences.

Lines 459-463: Odd expression, hard to understand.

Line 469-470: The antennal development does not provide sufficient evidence to illustrate the relationship between mantodea and Blattodea or Orthoptera. It only suggests such relationship.

Line 472-482: Unnecessary background information, these points have been given in the Introduction already. I wouldn’t emphasize too much on the function of sensilla, this is after all, a morphological paper. These functional speculations are not the main aims of this paper.

Line 503 paragraph: why not discuss the comparison with other mantids before with Blattodea (Line 483 paragraph)?

Author Response

Reviewer #2:

This version is much better than the original one, reflecting the input of the authors in improving this paper. I am happy for the authors to go ahead without my involvement. Apologies on my comments about the Tukey’s test, however, I would suggest either improve the table aesthetically, or turn it into a figure to better make your points. Table 1 at this state is a bit hard to read. Specific points:

Response:

Thank you again for your detailed revisions of our manuscript. The content has been modified based on the feedback received. Regarding the comment on improving the readability of Table 1, we fully agree with the reviewer’s observation. To address this, we has refined the table’s aesthetic design by reorganizing its structure, adjusting formatting. We also thank the reviewer for their gracious clarification regarding the earlier comments on Tukey’s test. We have carefully reviewed the methodology and confirmed its appropriateness for the study’s design.

Comment 1: Line 425. This is not substantiated. This paper only classified the sensilla by morphology, but numerous previous studies shown that functionally different sensilla not necessarily show morphological differences.

Response:

We agree that functionally different sensilla not necessarily show morphological differences. This sentence has been deleted in our manuscript, and the discussion section 7.1 has been rewrote. The specific content is modified as follows: We first introduce the position of the diversity of sensilla types in mantis among insects. Then, the classification basis and distribution density of these diversity sensilla are briefly described.

Comment 2: Lines 459-463: Odd expression, hard to understand.

Response:

We agree with this comment. We changed “In contrast, the number of new flagellomeres formed from the first flagellomere is irregular in H. patellifera antennae, but has the same way to add the flagellomeres at this point. However, the first flagellomere consistently generates a single new meristal segment at each molt in grasshoppers and mantophasmids. In addition, the H. patellifera antennae have a second site to add the flagellomeres” to “In contrast, in the antennae of H. patellifera, the number of newly formed flagellomeres from the first flagellomere is irregular, but the manner in which flagellomeres are added at this position is the same. However, in grasshoppers and mantophasmids, the first flagellomere consistently generates a new meristal segment with each molt. Compared to termites and cockroaches, the H. patellifera antennae have a second site to add the flagellomeres”.

Comment 3: Line 469-470: The antennal development does not provide sufficient evidence to illustrate the relationship between mantodea and Blattodea or Orthoptera. It only suggests such relationship.

Response:

We agree with this comment. We changed “Antennal development in insects illustrates that mantids have a closer relationship with termites and cockroaches, whereas they are more distantly related to Orthopteran species” to “Antennal development patterns further suggest that mantid is closely related to termites and cockroaches antennae”.

Comment 4: Line 472-482: Unnecessary background information, these points have been given in the Introduction already. I wouldn’t emphasize too much on the function of sensilla, this is after all, a morphological paper. These functional speculations are not the main aims of this paper.

Response:

We agree with this comment, according to your advice, we have deleted this paragraph.

Comment 5: Line 503 paragraph: why not discuss the comparison with other mantids before with Blattodea (Line 483 paragraph)?

Response:

We agree with this comment, according to your advice, we have reversed the order of the two sections in Section 7.3 ("Discussion").

Reviewer 3 Report

Comments and Suggestions for Authors

Dear Authors,

  1. Regarding "Sensilla Chaetica". I don't see any real difference between the subtypes. The authors don't provide dimensions of the sensilla. The length can be challenging to measure, but the diameter (at the base and middle) is easy to check.
    I have the impression that the authors are basing their findings on photographs taken at different scales. In addition, the sensilla have different positions, which makes them look different. But that doesn't mean they aren't the same.
  2. Regarding "Sensilla Trichodea". I still don't see the difference between StI and StII. Maybe there is, but authors can't prove it.
  3. Regarding "Sensilla Coelocapitula". Are the authors certain that these are indeed sensilla? I have looked as closely as the quality of the photos allows, and it appears that some of them (Sci II) resemble vertically arranged "Sensilla Grooved Peg".

In summary. The section "Types and Morphology of Sensilla in the Nymphs and Adults" is still unacceptable. The authors provide sources for terminology and characteristics of sensilla but do not adhere to them. Chapter "7.1 The Diversity of Antennal Sensilla" adds nothing to the manuscript and is a statement of the obvious. In some cases, it is actually difficult to determine the actual type of sensilla because there are no photos of them in large magnifications, and the images sent are of poor quality (1200 x 1100 pixels).

Author Response

Thank you again for your careful reading and invaluable comments, we have conscientiously revised our manuscript following your advice. Specific modifications are as follows.

Comment 1: Regarding "Sensilla Chaetica". I don't see any real difference between the subtypes. The authors don't provide dimensions of the sensilla. The length can be challenging to measure, but the diameter (at the base and middle) is easy to check.
I have the impression that the authors are basing their findings on photographs taken at different scales. In addition, the sensilla have different positions, which makes them look different. But that doesn't mean they aren't the same.

Response:

The classification of these three sensilla types is mainly based on length and their distribution. The length of the three subtypes of sensilla have been previously measured in all nymphs and adults, and found significant differences in their length. Taking the length of male adult as an example, the length of ScI vary greatly, about 26.84-125.05μm. The length of ScII is moderate, about 65.73-81.72μm. The length of ScIII is the shortest, about 46.99-51.55μm. The same is true for other instars and female adults. (Length measurements are submitted in the attachment). In terms of the distribution of sensilla chaetic, ScI is distributed in all flagellomeres, ScII is usually distributed in the middle of two ScI of the distal flagellomere, and ScIII is often distributed in the end of the proximal flagellomeres. In addition, there are certain differences in their morphology. Also taking adult male as an example, ScI was slender, and the width of the base of ScI was about 3.16-6.2μm. The width of the base of ScII was wider than that of ScI, about 5.96-6.36μm. The base of ScIII was the widest, about 6.45-6.76μm. The same is also true for other instars and female adults. (Width measurements are also submitted in the attachment)

Comment 2: Regarding "Sensilla Trichodea". I still don't see the difference between StI and StII. Maybe there is, but authors can't prove it.

Response:

The classification of these two sensilla types is mainly based on length and their distribution. The length of sensilla trichoidea I and II differed greatly. The length of sensilla trichoidea I is about 30~50um, while the length of sensilla trichoidea I is about 20~30um (Length measurements are also submitted in the attachment). In addition, the position of sensilla trichoidea was also different, sensilla trichoidea I distributed in almost all the flagellomeres and sensilla trichoidea II mainly distributed in the distal flagellomeres. In this paper, the classification of sensilla trichoidea was based on Carel's literature “The antennal sensilla of the praying mantis Tenodera aridifolia: a new flagellar partition based on the antennal macro-, micro- and ultrastructures”. They also classified sensilla trichoidea based on the length and their distribution, and sensilla trichoidea are divided into three subtypes.

Comment 3: I Regarding "Sensilla Coelocapitula". Are the authors certain that these are indeed sensilla? I have looked as closely as the quality of the photos allows, and it appears that some of them (ScoII) resemble vertically arranged "Sensilla Grooved Peg".

Response:

We agree with this comment. We sincerely apologize for this fault. After our inspection, factually, the “ScoII” in our manuscript is vertically arranged sensilla grooved peg. The primary reason for this misclassification was that during the experimental observation, these sensilla was in an upright state, and there was a certain gap in its bottom structure, which led to the mistaken belief that it grew in a cavity, so it was classified into sensilla coelocapitula. After examination, it was found that there are indeed small gaps in the bottom of these sensilla. We thought that these gap structures were more likely caused by antennal movement. Now, we corrected our mistake.

Comment 4: In summary. The section "Types and Morphology of Sensilla in the Nymphs and Adults" is still unacceptable. The authors provide sources for terminology and characteristics of sensilla but do not adhere to them. Chapter "7.1 The Diversity of Antennal Sensilla" adds nothing to the manuscript and is a statement of the obvious. In some cases, it is actually difficult to determine the actual type of sensilla because there are no photos of them in large magnifications, and the images sent are of poor quality (1200 x 1100 pixels).

Response:

Antennal sensilla have been identified and classified based on their external appearance, length and position in this study. The diversity of antennal sensilla is similar to T. aridifolia. But Sc is divided into three subtypes based on the length and position in H. patellifera. The specific discussion has been answered in the above (Comments 1-3).

We agree that the original version of Section 7.1 lacked interpretive depth. The specific content is modified as follows: We first introduce the position of the diversity of sensilla types in mantis among insects. Then, the classification basis and distribution density of these diversity sensilla are briefly described.

We apologize for the insufficient resolution of the original figures, which hindered accurate morphological assessment. All critical sensilla types are now represented in high-magnification SEM images.

Round 3

Reviewer 3 Report

Comments and Suggestions for Authors

Dear Authors,

The manuscript has been slightly revised. In my opinion, it's still not enough. I will not repeat here the words I wrote in previous reviews, which were not taken into account by the authors. I leave the decision on the manuscript to the editor.